# Plasma proteomic signatures of social isolation and loneliness associated with morbidity and mortality

Chun Shen[1,2,3], Ruohan Zhang[4], Jintai Yu [5], Barbara J. Sahakian [1,6,7] ✉, Wei Cheng [1,2,5] ✉ & Jianfeng Feng [1,2,4,8,9] ✉

The biology underlying the connection between social relationships and health is largely unknown. Here, leveraging data from 42,062 participants across 2,920 plasma proteins in the UK Biobank, we characterized the proteomic signatures of social isolation and loneliness through proteome-wide association study and protein co-expression network analysis. Proteins linked to these constructs were implicated in inflammation, antiviral responses and complement systems. More than half of these proteins were prospectively linked to cardiovascular disease, type 2 diabetes, stroke and mortality during a 14 year follow-up. Moreover, Mendelian randomization (MR) analysis suggested causal relationships from loneliness to five proteins, with two proteins (ADM and ASGR1) further supported by colocalization. These MR-identified proteins (GFRA1, ADM, FABP4, TNFRSF10A and ASGR1) exhibited broad associations with other blood biomarkers, as well as volumes in brain regions involved in interoception and emotional and social processes. Finally, the MR-identified proteins partly mediated the relationship between loneliness and cardiovascular diseases, stroke and mortality. The exploration of the peripheral physiology through which social relationships influence morbidity and mortality is timely and has potential implications for public health.

Social relationships are adaptive and critical for wellbeing and survival in social species[1]. Social isolation and loneliness, characterized as reflections of objective and subjective manifestations of impoverished social relationships, are increasingly recognized as important global public concerns[2]. Cumulative evidence demonstrates that both social isolation and loneliness are linked to morbidity and mortality, with effects comparable to traditional risk factors such as smoking and obesity[3–6]. Despite these empirical associations, the underlying mechanisms through which social relationships impact health remain elusive.

Experimental studies show that social interactions can causally alter animal physiology, such as sympathetic nervous system (SNS) and hypothalamic–pituitary–adrenal (HPA) activity, inflammation and antiviral responses, and directly influence disease risk[7–9]. These patterns parallel observations in human correlational studies[10,11]. Moreover, adverse social relationships have been associated with unhealthy

---

[1]Institute of Science and Technology for Brain-Inspired Intelligence, Fudan University, Shanghai, China. [2]Key Laboratory of Computational Neuroscience and Brain-Inspired Intelligence (Fudan University), Ministry of Education, Shanghai, China. [3]Department of Clinical Neurosciences, University of Cambridge, Cambridge, UK. [4]Department of Computer Science, University of Warwick, Coventry, UK. [5]Department of Neurology and National Center for Neurological Disorders, Huashan Hospital, State Key Laboratory of Medical Neurobiology and MOE Frontiers Center for Brain Science, Fudan University, Shanghai, China. [6]Department of Psychiatry, University of Cambridge, Cambridge, UK. [7]Behavioural and Clinical Neuroscience Institute, University of Cambridge, Cambridge, UK. [8]Zhangjiang Fudan International Innovation Center, Shanghai, China. [9]School of Data Science, Fudan University, Shanghai, China. ✉e-mail: bjs1001@cam.ac.uk; wcheng@fudan.edu.cn; jffeng@fudan.edu.cn

**Table 1 | Characteristics of study population at baseline in the UK Biobank**

| | All (*N*=42,062) | Social isolation at baseline | | | Loneliness at baseline | | |
|---|---|---|---|---|---|---|---|
| | | Not isolated (*N*=38,157) | Socially isolated (*N*=3,905) | *P* value | Not lonely (*N*=39,373) | Lonely (*N*=2,689) | *P* value |
| **Age at baseline, in years** | 56.4 (8.2) | 56.4 (8.2) | 56.7 (7.9) | 0.014 | 56.4 (8.2) | 56.2 (8.1) | 0.140 |
| **Sex** | | | | 0.426 | | | 0.812 |
| Female | 21,986 (52.3%) | 19,969 (52.3%) | 2,017 (51.7%) | | 20,574 (52.3%) | 1,412 (52.5%) | |
| Male | 20,076 (47.7%) | 18,188 (47.7%) | 1,888 (48.3%) | | 18,799 (47.7%) | 1,277 (47.5%) | |
| **Ethnicity** | | | | $2.9 \times 10^{-7}$ | | | 0.001 |
| White | 38,274 (91.0%) | 34,814 (91.2%) | 3,460 (88.6%) | | 35,887 (91.1%) | 2,387 (88.8%) | |
| Mixed | 1,718 (4.1%) | 1,535 (4.0%) | 183 (4.7%) | | 1,580 (4.0%) | 138 (5.1%) | |
| Asian | 1,503 (3.6%) | 1,320 (3.5%) | 183 (4.7%) | | 1,387 (3.5%) | 116 (4.3%) | |
| Black | 188 (0.4%) | 162 (0.4%) | 26 (0.7%) | | 174 (0.4%) | 14 (0.5%) | |
| Other | 379 (0.9%) | 326 (0.9%) | 53 (1.4%) | | 345 (0.9%) | 34 (1.3%) | |
| **Education level** | | | | $<2.2 \times 10^{-16}$ | | | $<2.2 \times 10^{-16}$ |
| College/university degree | 14,816 (35.2%) | 13,574 (35.6%) | 1,242 (31.8%) | | 14,129 (35.9%) | 687 (25.5%) | |
| Education to age 18 years | 14,083 (33.5%) | 12,878 (33.8%) | 1,205 (30.8%) | | 13,169 (33.4%) | 914 (34.0%) | |
| Education to age 16 years | 6,863 (16.3%) | 6,271 (16.4%) | 592 (15.2%) | | 6,372 (16.2%) | 491 (18.3%) | |
| No qualifications | 6,300 (15.0%) | 5,434 (14.2%) | 866 (22.2%) | | 5,703 (14.5%) | 597 (22.2%) | |
| **Household income** | | | | $<2.2 \times 10^{-16}$ | | | $<2.2 \times 10^{-16}$ |
| At least £31,000 | 21,334 (50.7%) | 20,107 (52.7%) | 1,227 (31.4%) | | 20,392 (51.8%) | 942 (35.0%) | |
| Less than £31,000 | 20,728 (49.3%) | 18,050 (47.3%) | 2,678 (68.6%) | | 18,981 (48.2%) | 1,747 (65.0%) | |
| **Current smoker** | | | | $<2.2 \times 10^{-16}$ | | | $<2.2 \times 10^{-16}$ |
| Yes | 4,450 (10.6%) | 3,721 (9.8%) | 729 (18.7%) | | 3,974 (10.1%) | 476 (17.7%) | |
| No | 37,612 (89.4%) | 34,436 (90.2%) | 3,176 (81.3%) | | 35,399 (89.9%) | 2,213 (82.3%) | |
| **Alcohol intake** | | | | $<2.2 \times 10^{-16}$ | | | $<2.2 \times 10^{-16}$ |
| At least three times per week | 18,868 (44.9%) | 17,556 (46.0%) | 1,312 (33.6%) | | 17,886 (45.4%) | 982 (36.5%) | |
| Twice or less per week | 23,194 (55.1%) | 20,601 (54.0%) | 2,593 (66.4%) | | 21,487 (54.6%) | 1,707 (63.5%) | |
| **Body mass index, kg m$^{-2}$** | 27.4 (4.8) | 27.4 (4.7) | 27.9 (5.3) | $6.0 \times 10^{-8}$ | 27.4 (4.7) | 28.5 (5.5) | $<2.2 \times 10^{-16}$ |

Two-sided *P* values were estimated by a *t*-test for continuous variables and chi-squared test for categorical variables.

lifestyles[12], potentially impacting these physiological pathways and subsequently affecting health. In comparison with behavioural moderators that indirectly influence health, there is a growing focus on understanding biological processes mediating the link between social relationships and health, given their relevance to improving disease prediction, prevention and intervention. It is noteworthy that proteins, as the final products of gene expression, serve as the main functional components of biological processes and represent a major source of drug targets[13]. Therefore, understanding the proteomic associations of social isolation and loneliness becomes imperative for unravelling the biology underpinning the effects of social relationships on health. One study reported that circulating brain-derived neurotrophic factor levels were associated with social relationships and partly mediated the effect between social support and dementia risk[14]. However, no comprehensive proteome-wide association study (PWAS) of social isolation and loneliness has been performed so far.

Here, leveraging high-throughput, population-scale proteomics alongside deep phenotypic data from the UK Biobank, we aimed, in this novel study, to answer two key questions: (1) What are the proteomic profiles associated with social isolation and loneliness? (2) How do proteomic alterations contribute to the relationship of social isolation, loneliness and health? To address the first question, we initially conducted PWASs and protein co-expression network analysis for social isolation and loneliness, respectively. The identified proteins and protein modules were subsequently examined potential causal

relationships with social isolation and loneliness using bidirectional Mendelian randomization (MR) and colocalization analyses. To explore the relationship between the MR-identified proteins and broad physical functions, we investigated their associations with other blood biomarkers. On the basis of the social brain hypothesis[15] and increasing research on the neurobiology of social isolation and loneliness[16], we further related these proteins to brain volumes. For the second question, we delved into the prospective associations between proteins linked to social isolation and loneliness and the incidence of morbidity and mortality. Specifically, we focused on five diseases with well-documented associations with social relationships: cardiovascular diseases (CVDs)[17,18], dementia[19,20], type 2 diabetes (T2D)[21], depression[22] and stroke[18,23]. Finally, we evaluated the role of plasma proteins in the connection between social relationships and the risk of morbidity and mortality in mediation analyses for time-to-event outcomes.

## Results

### Cohort characteristics

Our primary study population included 42,062 participants (aged 56.4 ± 8.2 years, 52.3% female) from the UK Biobank who had quality-controlled proteomic data and complete behavioural data including social isolation, loneliness and all covariates. A flow chart of the participant selection is shown in Supplementary Fig. 1. Among these, 3,905 (9.3%) reported being socially isolated and 2,689 (6.4%) felt lonely. Detailed demographic characteristics, stratified by social

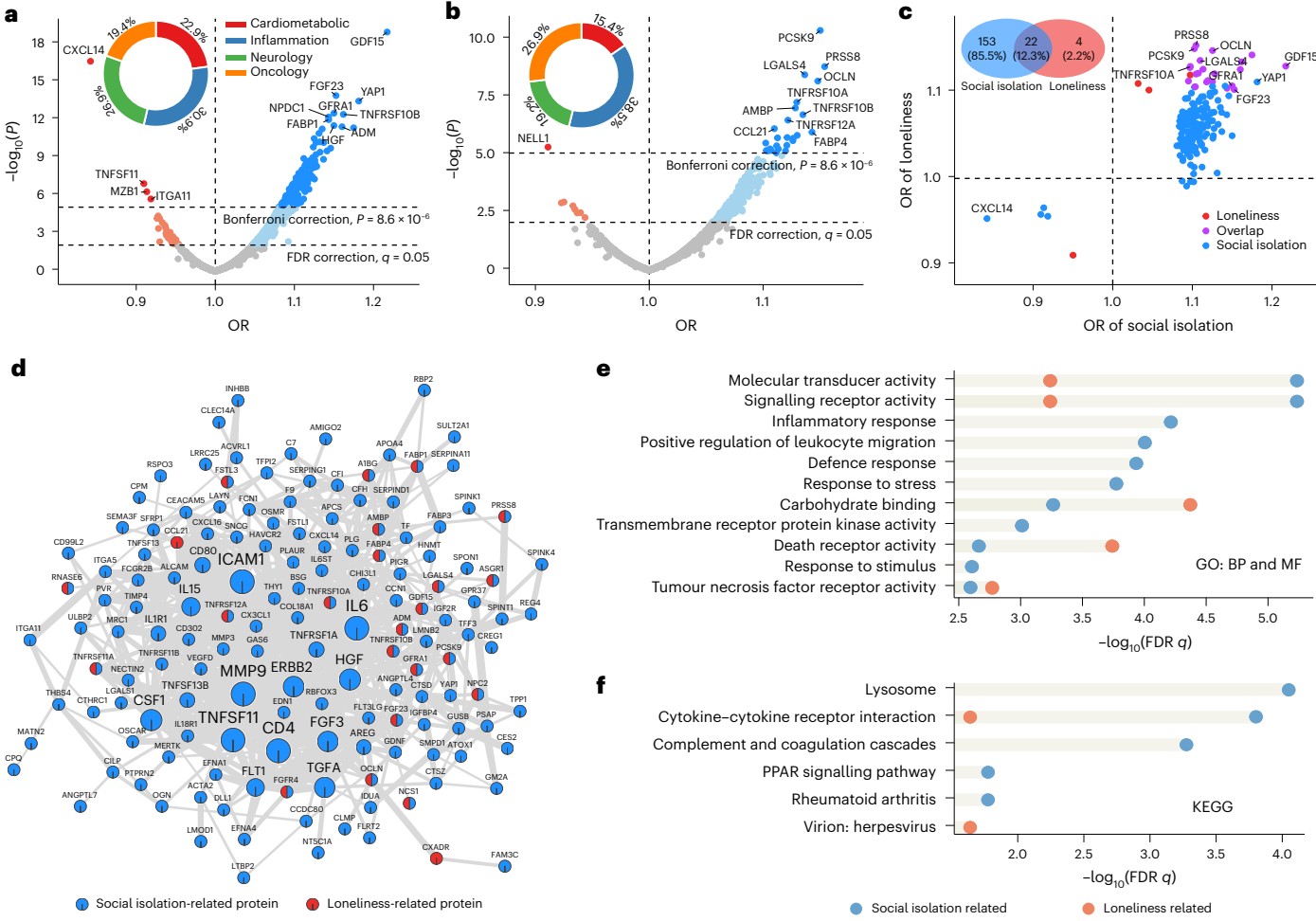

**Fig. 1 | PWAS for social isolation and loneliness.** The average sample size for the 2,920 proteins analysed is 37,704, ranging from 30,778 to 41,396. Sample sizes for proteins significantly associated with social isolation and loneliness after Bonferroni correction ($P < 0.05/(2,920 \times 2) = 8.6 \times 10^{-6}$) are detailed in Supplementary Tables 1 and 2. Logistic regression models were adjusted for age, sex, site, batch, time gap between blood collection and protein measurement, ethnicity, education level, household income, smoking, alcohol consumption, BMI and the first 20 genetic PCs. All statistical tests were two sided. **a**, A volcano plot displaying the correlation of protein abundance with social isolation. The $x$ axis represents ORs, and the $y$ axis represents $-\log_{10}(P$ values). The dashed lines indicate the thresholds for Bonferroni and FDR corrections ($q < 0.05$) when considering social isolation and loneliness simultaneously. The pie charts depict the proportions of identified proteins across four distinct panels.

**b**, A volcano plot displaying the correlation of protein abundance with loneliness. **c**, A scatter plot visualizing the 179 proteins significant after Bonferroni correction. The $x$ axis represents the ORs from the PWAS for social isolation and the $y$ axis represents the ORs from the PWAS for loneliness. The Venn diagram shows the overlap of proteins associated with social isolation or loneliness. **d**, A PPI network for the 179 identified proteins. The node size reflects the modularity level estimated by the MCC method and line thickness represents the interaction score. **e**, The top five enriched GO biological processes (BP) and molecular functions (MF) for proteins related to social isolation and loneliness. The $x$ axis represents $-\log_{10}$(FDR $q$ values). Functional enrichments that are also related to proteins associated with the other construct are marked. **f**, The top five enriched KEGG pathways for proteins related to social isolation and loneliness. PPAR, peroxisome proliferator-activated receptor.

isolation and loneliness, are presented in Table 1. During a median (s.d.) follow-up of 13.7 (2.1) years (ending on 30 November 2022 or death), 2,695 participants developed CVD, 892 developed all-cause dementia, 1,703 developed T2D, 1,521 developed depression, 983 developed stroke and 4,255 passed away.

## Proteins associated with social isolation and loneliness

We conducted logistic regression for PWASs involving 2,920 plasma proteins, using social isolation or loneliness as the outcome. In simple models incorporating age, sex, site, technical factors and the first 20 genetic principal components (PCs) as covariates, we found 776 proteins significantly associated with social isolation and 519 proteins associated with loneliness ($P < 0.05/(2,920 \times 2) = 8.6 \times 10^{-6}$) (Supplementary Fig. 2). After additional adjustments for ethnicity, education level, household income, smoking, alcohol consumption and body mass index (BMI), 175 proteins associated with social isolation (Fig. 1a

and Supplementary Table 1) and 26 proteins associated with loneliness (Fig. 1b and Supplementary Table 2) maintained significance at the Bonferroni-corrected threshold.

Notably, growth differentiation factor 15 (GDF15), a protein belonging to the transforming growth factor-β superfamily that acts as an inflammatory marker[24], demonstrated the strongest association with social isolation (odds ratio (OR) of 1.22, 95% confidence interval (CI) 1.17 to 1.27, $P = 1.2 \times 10^{-19}$, $N = 41,396$), while proprotein convertase subtilisin/kexin type 9 (PCSK9), a key protein in the regulation of cholesterol metabolism[25], showed the strongest association with loneliness (OR of 1.15, 95% CI 1.10 to 1.20, $P = 4.2 \times 10^{-11}$, $N = 41,396$). The majority of identified proteins exhibited a positive association, indicating that higher protein abundance was linked to an elevated risk of social isolation or loneliness. However, only four proteins emerged as protective factors against social isolation, and one against loneliness. Particularly notable among these was C-X-C motif chemokine ligand-14 (CXCL14),

an immune and inflammatory modulator[26], emerging as the second most significant protein associated with social isolation (OR of 0.84, 95% CI 0.81 to 0.88, $P = 2.4 \times 10^{-17}$, $N = 40,090$).

Furthermore, 22 proteins (12.3% of the overall identified proteins) were common (Fig. 1c), and the proteome-wide associative patterns of social isolation and loneliness were moderately related ($r = 0.54$, 95% CI 0.52 to 0.57, $P < 2.2 \times 10^{-16}$, $N = 2,920$; Supplementary Fig. 3), implicating shared and distinct proteomic signatures between social isolation and loneliness. Restricted cubic splines[27] revealed that four proteins displayed a significant nonlinear association with social isolation ($P < 0.05/(175 + 26) = 2.5 \times 10^{-4}$; Supplementary Fig. 4), while no nonlinear proteomic association with loneliness was detected.

### Sensitivity analyses

First, ordered logistic regressions using raw scores of social isolation and loneliness were performed to test the robustness and potential dose-dependent associations with plasma protein levels. This approach yielded results consistent with those from dichotomous variables and logistic regression but with greater statistical power (580 and 125 proteins associated with social isolation and loneliness, respectively; Supplementary Fig. 5).

Given the reported sampling bias in the UK Biobank Pharma Proteomics Project (UKB-PPP) subcohort composition, we replicated the primary analyses specifically within the randomly selected subset ($N = 36,250$), which is highly representative of the overall UK Biobank population. The proteins identified in this subset were consistent with those in all available samples and the proteomic associative patterns in these two populations were highly correlated (both $r > 0.9$, $P < 2.2 \times 10^{-16}$; Supplementary Fig. 6). Additionally, to mitigate potential population stratification, we conducted PWAS specifically in Caucasians ($N = 35,697$) and observed that the proteomic associative patterns were highly correlated with those found in the full sample (both $r > 0.9$, $P < 2.2 \times 10^{-16}$; Supplementary Fig. 7).

Next, interaction terms between sex or age and protein levels were included in logistic models to assess potential sex and age differences. No significant interaction effects between sex or age and proteins associated with social isolation or loneliness were observed ($P > 0.05/(175 + 26) = 2.5 \times 10^{-4}$). The PWAS results for males ($N = 20,076$) and females ($N = 21,986$) are shown in Supplementary Fig. 8. Additionally, the PWAS results for younger (<60 years, $N = 24,171$) and older (≥60 years, $N = 17,891$) groups are shown in Supplementary Fig. 9.

Considering the established link between loneliness and depression[22], we explored the potential influence of depressive symptoms on the proteomic association of social isolation and loneliness ($N = 38,778$). The adjustment for depressive symptoms modestly impacted the association between social isolation and proteins (Supplementary Fig. 10a,b), whereas notably weakening the association between loneliness and proteins (for example, the OR of GDF15 decreased by 9.5% after adjustment; Supplementary Fig. 10c–e). Additionally, we examined the potential influence of physical activity ($N = 34,548$) and found that the associations between social isolation, loneliness and plasma proteins were largely independent of physical activity levels (Supplementary Fig. 11).

Incorporating both social isolation and loneliness into the model had minimal impact on their respective associations with proteins (Supplementary Fig. 12). To further investigate the underlying construct of impoverished social relationships, multinomial logistic regressions were conducted to test the association between the four-group classification and plasma protein levels. Compared with participants who were neither isolated nor lonely ($N = 36,100$), 116 proteins differed significantly in the socially isolated (SI) but not lonely (LO) group (SI+LO−; $N = 3,273$), 8 in the not isolated but lonely group (SI−LO+; $N = 2,057$) and 22 in the socially isolated and lonely group (SI+LO+; $N = 632$) (Supplementary Fig. 13). Interestingly, GDF15 was identified

as the top differentiated protein in SI+LO− and SI+LO+, while PCSK9 was the top differentiated protein in SI−LO+.

Finally, we performed cross-validation by randomly splitting the UK Biobank samples 100 times. Our results showed that most of the significant proteins identified using the full sample retained significance in at least one of the two split samples, and proteomic associative patterns for social isolation (mean $r = 0.56$, s.d. 0.05) and loneliness (mean $r = 0.35$, s.d. 0.08) between the two split samples exhibited medium to large correlations (Supplementary Fig. 14).

### PPI and pathway enrichment

To explore potential interactions among proteins associated with social isolation or loneliness, we analysed the protein–protein interaction (PPI) networks within the identified pool of 179 proteins using the STRING database[28]. A prominent interconnected network featuring 150 proteins and 690 PPIs was found (Fig. 1d). The maximal clique centrality (MCC) method[29] revealed the top three hub proteins of interleukin 6 (IL6), intercellular adhesion molecule-1 (ICAM1) and cluster of differentiation 4 (CD4) all exhibiting associations with social isolation.

Furthermore, we investigated functional pathways for the proteins associated with social isolation and loneliness at a false discovery rate (FDR) $q < 0.05$. We observed several shared pathways between social isolation and loneliness. Both sets of proteins exhibited significant enrichment in immune pathways such as cytokine–cytokine receptor interaction and antiviral processes (Fig. 1e,f and Supplementary Tables 3 and 4). Additionally, proteins associated with social isolation specifically showed enrichment in complement system and mitogen-activated protein kinase (MAPK)/extracellular signal-regulated kinase (ERK) signalling.

### Protein networks linked to social isolation and loneliness

Given that complex diseases are not caused by individual proteins, but rather result from highly interactive protein networks[30], we applied a data-driven approach to classify plasma proteins into clusters or modules based on protein co-expression patterns. Thirteen non-overlapping protein modules were identified, ranging in size between 30 and 1,051 proteins (Supplementary Fig. 15). We then correlated the module eigengene with social isolation and loneliness, revealing that M4, M8 and M12 showed associations with both conditions after Bonferroni correction ($P < 0.05/(13 \times 2) = 0.002$) (Fig. 2a,b). Additionally, M3 was found to be associated with social isolation. Enrichment analyses suggested distinct functions associated with these modules. For example, protein module M4 exhibited enrichment in immune-related pathways (Fig. 2c) and included the top protein related to social isolation, GDF15 (Supplementary Table 5). M8 demonstrated enrichment in metabolic processes (Fig. 2d and Supplementary Table 6). M12 showed enrichment in neutrophil degranulation (Fig. 2e and Supplementary Table 7). M3 was enriched in complement systems (Fig. 2f) and included the top protein related to loneliness, PCSK9 (Supplementary Table 8). Actually, M3 was also associated with loneliness if relaxing the significance threshold to FDR correction. Sensitivity analysis demonstrated that the biological relevance of the identified modules associated with social isolation and loneliness remained robust across various co-expression network construction parameters (Supplementary Figs. 16–18).

Moreover, we examined the relationship between significant modules with the proteins identified by PWAS using two-sided Fisher's exact tests. After Bonferroni correction, both protein sets associated with social isolation and loneliness were significantly enriched in M4, and the results were consistent across all proteins, the top 20% and the top 10% of proteins in each module (Supplementary Table 9). It is noteworthy that both approaches—the PWAS and the protein co-expression networks—were consistent and complementary, together providing more comprehensive information.

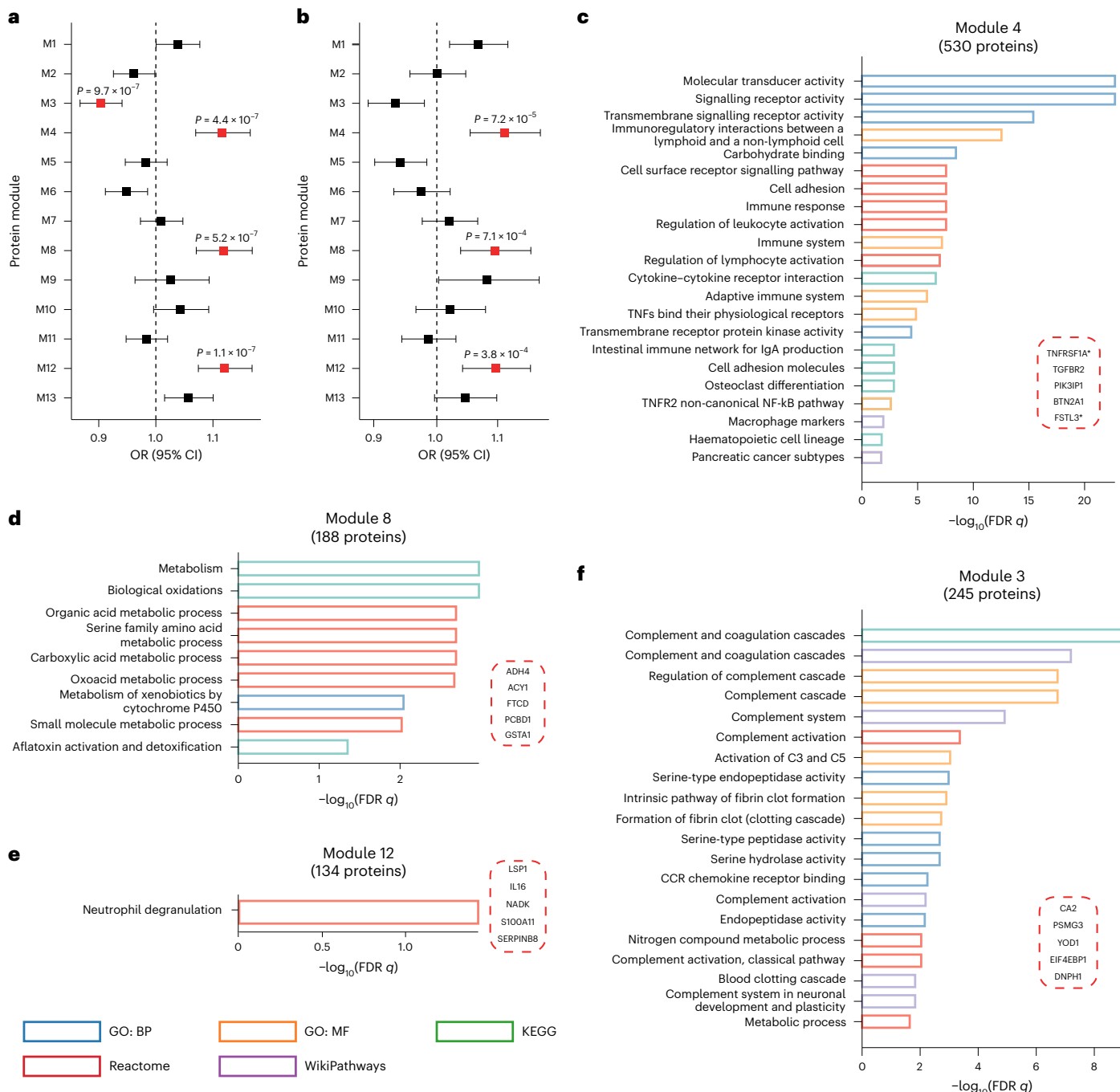

**Fig. 2 | Association of protein co-expression network with social isolation and loneliness (N = 35,475). a**, A forest plot showing the associations between protein modules and social isolation. M1 to M13 represent 13 modules identified through protein network analysis. Two-sided P values were derived from logistic regression models adjusted for the same covariates as in the PWAS. The dots represent ORs, and error bars indicate 95% CIs. The red markers denote significance after Bonferroni correction (P < 0.05/26 = 0.002). **b**, A forest plot showing the associations between protein modules and loneliness.

**c**–**f**, Enrichment analysis of proteins from modules 4 (**c**), 8 (**d**), 12 (**e**) and 3 (**f**). The top five significantly enriched pathways from each database are displayed, with the x axis representing −log$_{10}$(FDR q values). The top five proteins in each module that are most highly correlated with the overall network expression are displayed in the dashed box. Red indicates a positive correlation, while blue indicates a negative correlation. Proteins marked with asterisks overlap with those identified in the PWAS. CCR, CC chemokine receptor.

## MR between social isolation, loneliness and proteins

To infer causality between social isolation, loneliness and the identified proteins and protein modules, we implemented a bidirectional two-sample MR. Genome-wide association study (GWAS) summary statistics were sourced from non-overlapping samples from the UK Biobank. In the forward direction, no protein or protein network was found to be associated with social isolation or loneliness at an FDR significance threshold, using either *cis*-protein quantitative trait loci (pQTLs) alone or a combination of *cis*- and *trans*-pQTLs (Supplementary Table 10). However, we uncovered significant associations between loneliness and five proteins in the backwards direction using the inverse-variance weighting (IVW) method (FDR q < 0.025) using both *cis*- and *trans*-pQTLs (Fig. 3 and Supplementary Table 11). The results of the sensitivity analyses were consistent with IVW estimates in direction

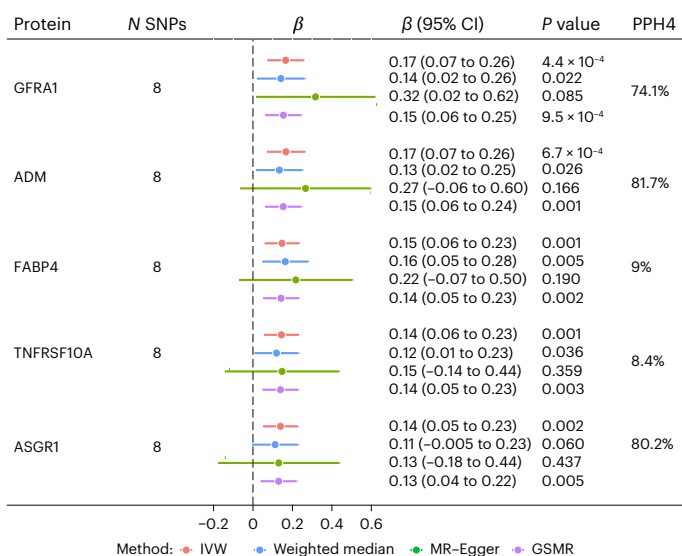

**Fig. 3 | Summary of MR and colocalization analysis on the associations between loneliness and proteins and protein modules.** The IVW method was used as the primary analytic approach, complemented by weighted median, MR–Egger and GSMR. The GWAS sample sizes were 297,396 for social isolation, 288,696 for loneliness, 35,327 for GFRA1, 35,385 for ADM, 35,544 for FABP4, 34,842 for TNFRSF10A and 34,842 for ASGR1. All statistical tests were two sided, and FDR correction was applied separately for each direction, as well as for social isolation and loneliness, with significance set at $q < 0.025$. Significant proteins identified by the IVW method are shown. The dots represent beta values and error bars indicate 95% CIs. Support for colocalization was considered strong for PPH4 $\geq 0.8$ and medium for $0.5 < \text{PPH4} < 0.8$.

and magnitude. No evidence of heterogeneity ($Q$ statistic, all $P > 0.3$) and horizontal pleiotropy (MR–Egger intercept, all $P > 0.3$; heterogeneity in independent instrument-outlier test, $P > 0.01$) among instrument variables (IVs) was detected. A leave-one-out analyses demonstrated no potentially influential single-nucleotide polymorphisms (SNPs) driving the causal link (Supplementary Fig. 19). However, the causal analysis using summary effect (CAUSE) method could not distinguish a model of causality from correlated pleiotropy for the five proteins that showed significant causal links with loneliness using IVW (Supplementary Table 12).

For the MR-identified proteins in relation to loneliness, we implemented colocalization analysis to ensure that the results were not confounded by linkage disequilibrium. Two proteins, ADM and asialoglycoprotein receptor 1 (ASGR1), exhibited strong evidence of colocalization (posterior probability of hypothesis 4 (PPH4) > 0.8 for one common causal variant) for at least one of the instruments, while GDNF family receptor alpha 1 (GFRA1) showed medium support for colocalization between loneliness and pQTL signals (PPH4 of 0.741) (Fig. 3).

**Broad phenotypic associations of MR-identified proteins**
We then extended our analysis to explore the broad associations of the MR-identified proteins with diverse blood biomarkers and brain volume. Following Bonferroni correction ($P < 0.05/(229 \times 5) = 4.4 \times 10^{-5}$), we observed extensive associations of these proteins with markers in biochemistry, haematology and metabolomics (Fig. 4a and Supplementary Table 13). Notably, cystatin C showed the strongest association with all of these five proteins. Additionally, all the MR-identified proteins and protein modules demonstrated robust connections with C-reactive protein (CRP) ($t = (13.2, 32.8)$, $N = (40,491, 41,145)$, $P < 9.6 \times 10^{-40}$), a substantial biomarker of systemic inflammation[31]. In brain level, four proteins were significantly associated with volumes in brain regions predominantly located in insula, caudate and frontal cortex, which are known to be involved in interoception, emotional and social

processes[32–34] (Fig. 4b and Supplementary Table 14). A higher level of ADM was related to lower grey matter volumes in bilateral insula and left caudate (Fig. 4c).

**Associations of related proteins with morbidity and mortality**
We examined the relationship of the 179 unique proteins and four protein modules linked to social isolation or loneliness with five diseases and mortality using Cox proportional hazards models. With adjustment of demographic, socioeconomical and lifestyle confounders and the first 20 genetic PCs, our findings revealed that 90.2% of these proteins were associated with mortality and over 50% were linked to T2D, CVD and stroke, whereas only 6.6% were associated with dementia ($P < 0.05/(183 \times 6) = 4.6 \times 10^{-5}$; Fig. 5a and Supplementary Table 15). Remarkably, GDF15 emerged as the top protein associated with an increased risk of all diseases and mortality, excluding T2D, while M8 demonstrated the strongest association with T2D (hazard ratio (HR) of 2.14, 95% CI 2.02 to 2.27, $P = 1.6 \times 10^{-146}$, $N = 36,534$). In the case of the MR-identified proteins, we observed significant associations with CVD, T2D, stroke and mortality. ADM was the only one showing an association with dementia (Fig. 5b).

**Mediating role of proteins in loneliness and health outcomes**
Finally, we delved into the potential mediating role of proteins, which have been implicated as causally linked to loneliness, in the relationship between loneliness and health outcomes. Initially, we estimated the percentage of excess risk mediated (PERM)[35,36] by each MR-identified protein for the association between loneliness and diseases and mortality through two Cox proportional models. After controlling for demographic, socioeconomical and lifestyle confounders, loneliness was not associated with the incidence of T2D and was therefore excluded from further mediation analyses. The largest effects were observed for mortality (PERM 8.6–16.3%) and CVD (PERM 5.6–8.3%) (Supplementary Table 16). Notably, ADM emerged as the primary mediator linking loneliness to various health outcomes, including CVD (8.3%), dementia (4.4%), stroke (7.8%) and mortality (16.3%). Although this approach provided a quantifiable contribution of proteins and is widely used in epidemiological research[37,38], the statistical significance and causal interpretation were unclear. Consequently, we used a counterfactual-based mediation analysis[39] to assess the direct and indirect effects in the relationship between loneliness and health outcomes through the MR-identified proteins. All five proteins significantly mediated the association between loneliness and CVD, stroke and mortality after Bonferroni correction (Fig. 5c and Supplementary Table 17). The estimated proportion of the indirect effect to the total effect was comparable to the PERM results. Interestingly, ADM was the only protein that significantly mediated the relationship between loneliness and all four diseases (CVD: indirect effect $9.0 \times 10^{-5}$, 95% CI $7 \times 10^{-5}$ to $1.1 \times 10^{-4}$, $P < 1 \times 10^{-16}$, $N = 37,846$; dementia: indirect effect $2.7 \times 10^{-5}$, 95% CI $1.6 \times 10^{-5}$ to $3.7 \times 10^{-5}$, $P = 1.4 \times 10^{-6}$, $N = 41,309$; depression: indirect effect $1.8 \times 10^{-5}$, 95% CI $7.1 \times 10^{-6}$ to $2.9 \times 10^{-5}$, $P = 0.001$, $N = 37,330$; and stroke: indirect effect $3.5 \times 10^{-5}$, 95% CI $2.5 \times 10^{-5}$ to $4.6 \times 10^{-5}$, $P = 5.4 \times 10^{-11}$, $N = 40,555$) and mortality (indirect effect $2.6 \times 10^{-4}$, 95% CI $2.3 \times 10^{-4}$ to $2.9 \times 10^{-4}$, $P < 1 \times 10^{-16}$, $N = 41,389$).

## Discussion
Understanding the proteomic signatures associated with social isolation and loneliness can provide insight into relevant biological processes, with implications for public health. Leveraging data from 2,920 plasma protein analytes across over 40,000 UK Biobank participants, the present study comprehensively characterized proteins and protein networks related to social isolation and loneliness. The identified proteins were highly interactive and were largely enriched in inflammation, antiviral responses and complement systems. Notably, more than half of these proteins were prospectively linked to CVD,

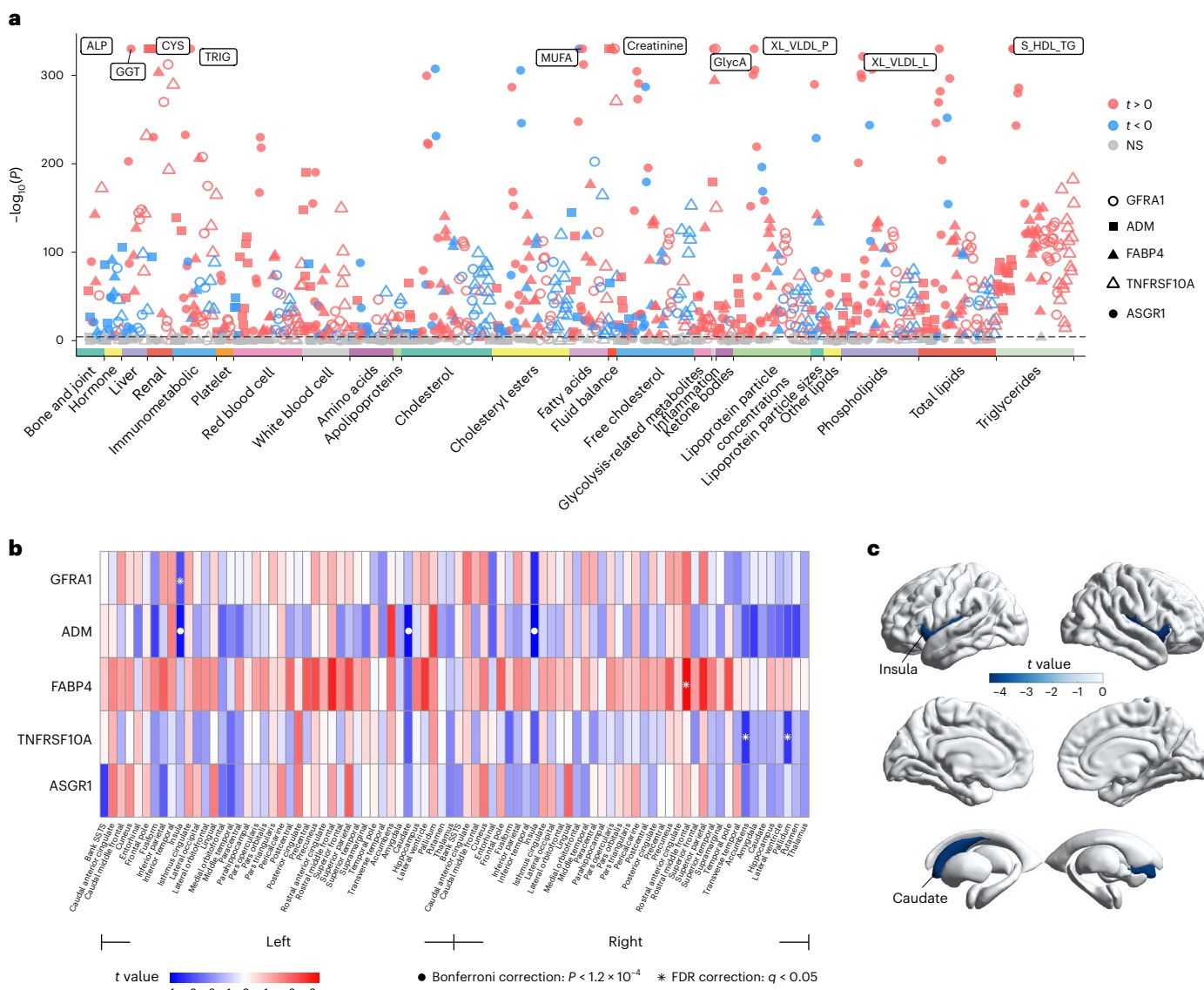

**Fig. 4 | Association of MR-identified proteins with blood biomarkers and brain volumes. a**, The association of MR-identified proteins with biochemical, haematological and metabolic biomarkers. The average sample size for the 229 blood markers analysed was 28,501, ranging from 3,696 to 42,145 (detailed in Supplementary Table 13). Two-sided $P$ values were obtained from linear regression models adjusted for the same covariates as in the PWAS. The $x$ axis represents various blood markers categorized into different groups. The symbols represent different proteins and protein modules, with colours indicating the type of correlation. The dashed line represents the Bonferroni correction ($P < 0.05/(229 \times 5) = 4.4 \times 10^{-5}$). **b**, The association of MR-identified proteins with cortical and subcortical volumes. The average sample size was 5,027, ranging from 4,982 to 5,085 (detailed in Supplementary Table 14). Two-sided $P$ values were derived from linear regression models adjusted for age, sex, imaging collection site, batch, time gap between blood collection

and protein measurement, ethnicity, education level, household income, smoking, alcohol consumption, BMI, the first 20 genetic PCs, intracranial volume and time gap between baseline and imaging collection. The colour bar represents $t$ values, the circles represent significance after Bonferroni correction ($P < 0.05/(84 \times 5) = 1.2 \times 10^{-4}$) and asterisks represent significance after FDR correction ($q < 0.05$) when considering all tested proteins and protein modules simultaneously. Detailed $P$ values are reported in Supplementary Table 14. **c**, Brain volumes significantly associated with the abundance of ADM after FDR correction. The colour bar represents $t$ values. ALP, alkaline phosphatase; GGT, gamma glutamyltransferase; CYS, cystatin C; TRIG, triglycerides; MUFA, monounsaturated fatty acids; GlycA, glycoprotein acetyls; XL_VLDL_P, concentration of very large very low-density lipoproteins (VLDL) particles; XL_VLDL_L, total lipids in very large VLDL; S_HDL_TG, triglycerides in small high-density lipoprotein.

T2D, stroke and mortality during a 14 year follow-up. Using MR, we suggested that loneliness causally contributed to the abundance of five proteins, with two proteins further supported by colocalization. These MR-identified proteins exhibited wide associations with biochemical, haematological and metabolic blood biomarkers, as well as volumes in brain regions involved in interoception and emotional and social processes. Finally, we showed that these MR-identified proteins partly mediated the relationship between loneliness and CVD, stroke and mortality. Collectively, these findings support the idea that social

relationships indirectly influence morbidity and mortality through peripheral physiological pathways.

Social isolation and loneliness, covering separate dimensions of social relationships, can operate independently. The reported behavioural correlations between these constructs range from small to medium ($r = 0.2$–$0.4$)[40,41]. They may impact health through distinct pathways, for instance, objective social engagement and loneliness exhibit different associations with neuroimmune markers in older age[42]. Our study revealed a moderate relationship in proteome-wide

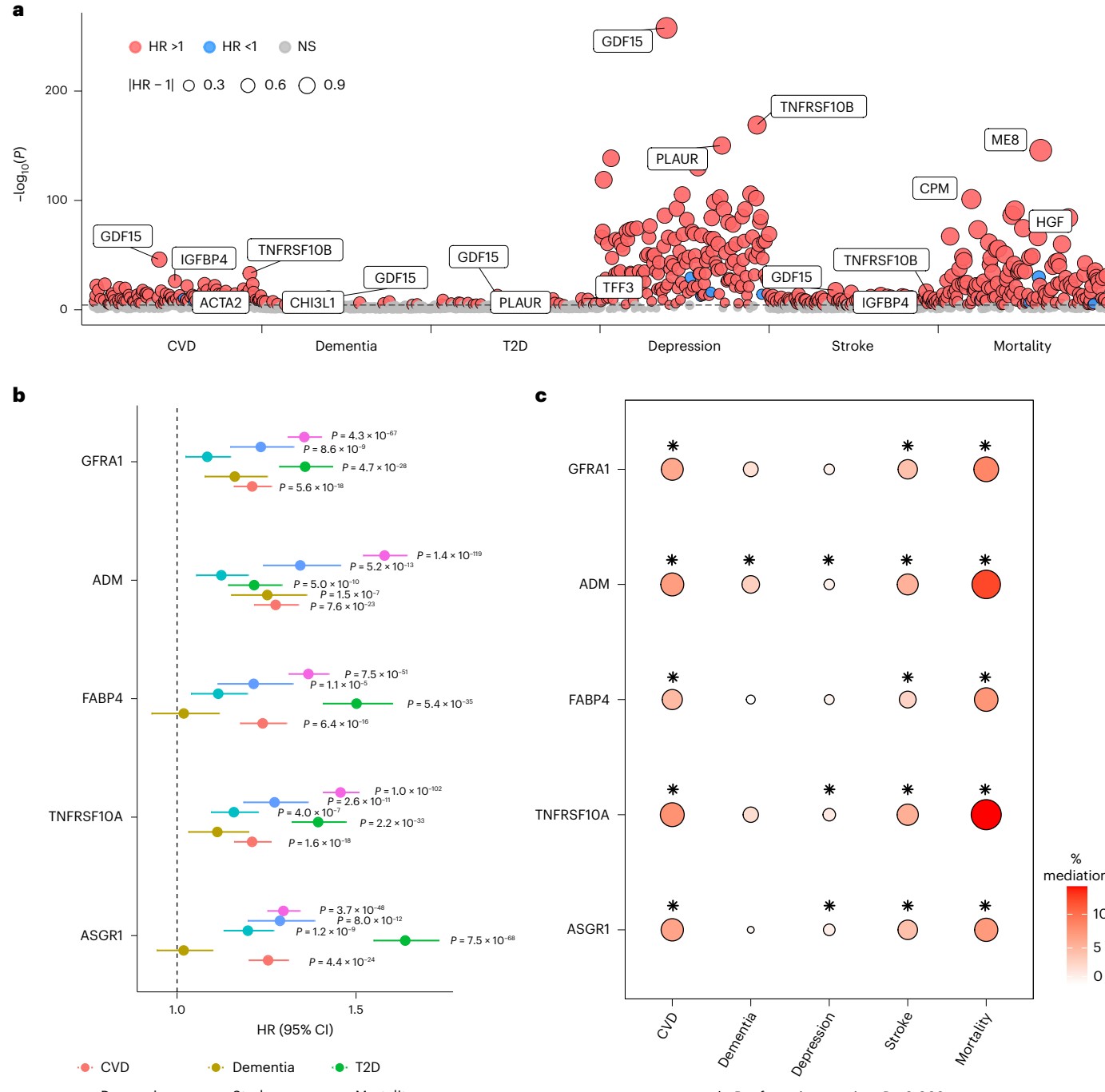

**Fig. 5 | Associations between identified proteins and morbidity and mortality.**
**a**, The associations between proteins and protein modules related to social isolation or loneliness with the incidence of five diseases and mortality. The average sample size was 37,808, ranging from 30,232 to 41,396 (detailed in Supplementary Table 15). Two-sided $P$ values were obtained from Cox proportional hazards models adjusted for age, sex, ethnicity, education level, household income, smoking, alcohol consumption, BMI, time gap between blood collection and protein measurement and the first 20 genetic PCs. The top three proteins most strongly associated with each disease and mortality are labelled. The dot colour indicates the type of correlation and the dot size represents the absolute difference between HR and 1. The dashed line represents the Bonferroni correction ($P < 0.05/(183 \times 6) = 4.6 \times 10^{-5}$). **b**, A forest plot showing the associations between the five MR-identified and health outcomes. The dots represent HRs, with error bars indicating 95% CIs. Significant $P$ values after Bonferroni correction are indicated. **c**, A mediation analysis of MR-identified proteins on the relationship between loneliness and five health outcomes using counterfactual-based analysis. The average sample size was 39,456, ranging from 36,735 to 41,571 (detailed in Supplementary Table 17). The dot size and colour represent the proportion of mediation estimated by indirect effect/(indirect effect + direct effect). The asterisks indicate significance after Bonferroni correction ($P < 0.05/(5 \times 5) = 0.002$). All statistical tests are two sided, and detailed $P$ values are reported in Supplementary Table 17.

associative patterns between social isolation and loneliness ($r = 0.54$). The quantity of proteins significantly linked to social isolation is six times higher than those associated with loneliness. Sensitivity analysis showed that additionally controlling for the other construct had a minimal impact on proteomic associations, suggesting independence in the proteomic association with each construct. However, there was substantial overlap, with approximately 85% of the proteins associated with loneliness being shared with social isolation. The identified

proteins associated with social isolation or loneliness formed a large interactive network, with immune-related proteins such as IL6 acting as hub proteins. Co-expression network analysis further confirmed shared and distinct modules associated with social isolation and loneliness.

The coregulation between inflammation and social behaviour has been widely discussed in the literature[43]. Social isolation and loneliness, acting as social stressors, can trigger stress responses, involving the SNS and HPA axis, leading to altered immune function[44]. Indeed, social isolation and loneliness have been shown to be associated with heightened proinflammatory activity, manifested by elevated levels of inflammation biomarkers such as IL6, fibrinogen and CRP[45–47]. In addition to inflammation, the SNS and HPA axis may also modulate antiviral responses. Older adults experiencing high levels of loneliness displayed an upregulation of proinflammatory genes along with a downregulation of antiviral genes, known as the conserved transcriptional response to adversity (CTRA)[48]. Mechanistic analyses in a macaque model of loneliness confirmed CTRA upregulation preceding increases in loneliness[49]. Another macaque study discovered that social isolation induced CTRA activation and the antiviral gene expression was impeded by enhancing prosocial engagement[50]. However, it should be noted that the majority of research exploring the relationship between social behaviour, antiviral processes and inflammation so far is based on small sample sizes and results are mixed due to methodological heterogeneity[10]. Our study, utilizing a biobank-level dataset and various analytical strategies, substantiated the association between social isolation, loneliness and proteins implicated in inflammation and antiviral processes.

Furthermore, both PWAS and network analysis revealed an association between social isolation and proteins linked to complement activation. Although no direct association was found between loneliness and complement in PWAS, loneliness was significantly linked to the protein M3 after FDR correction, which exhibited enrichment in the complement system. Complement acts as a rapid and efficient immune surveillance system, playing a pivotal role in maintaining homeostasis[51], a balance that could be disrupted by social adversity. Recent studies propose that protein components of the complement and coagulation systems could be key pathways in the pathology of psychosis as these pathways can become active years before the onset of psychosis, demonstrating sensitivity to the progression and manifestation of psychotic symptoms[52]. Moreover, we observed that social isolation was uniquely associated with MAPK/ERK signalling. Animal studies have shown that ERK2 conditional knockout mice displayed lower levels of social interactions[53]. Pair-housing could reverse the detrimental effects of post-stroke social isolation mediated by brain-derived neurotrophic factor via downstream MAPK/ERK signalling[54]. Both human and animal studies suggest that ERK is crucial for pathogenesis, symptomatology and treatment of depression[55].

Beyond the associative analyses, we integrated proteomic and genetic data to infer the directionality of the relationship between plasma proteins, social isolation and loneliness. Interestingly, we identified significant causal impacts only from loneliness to five proteins, and the robustness of these findings was confirmed through sensitivity analyses, except the CAUSE model. Prior research has established that protein–phenotype associations with both MR and colocalization evidence can predict a higher likelihood of success for a particular target-indication pair, making these proteins more likely to inform clinical applications[56]. In our study, the associations of loneliness with ADM and ASGR1 were strongly supported by colocalization. By examining correlations with other blood biomarkers in the UK Biobank, we discovered that both ADM and ASGR1 were strongly associated with a range of biochemical, haematological and metabolic biomarkers, including CRP, cholesterol and triglycerides. These findings are consistent with previous studies reporting the role of ADM in organizing neuroendocrine responses to stress, including simulating the SNS

and HPA axis[57], as well as regulating stress hormones[58]. Brain-derived ADM has the potential to control oxytocin release[59], which is a crucial social peptide[60]. Additionally, variant ASGR1 was associated with low cholesterol and a reduced risk of CVD[61,62]. We also explored the relationship between the MR-identified proteins and brain volumes, aligning with the growing research focus of characterizing neural mechanisms underlying social relationships[63], as related neural mechanisms may translate social experience into health-relevant physiological responses[44]. The strongest association was found between ADM and the insula, a hub for interoceptive mechanisms[64]. Interoception is proposed to function as a vital pathway for the brain–body interactions underlying the loneliness–health link[65]. Other significant associations were observed between plasma ADM levels and the left caudate, a region involved in emotional, reward and social processes[44,66].

While the link between social relationships and health is well established, uncovering the underlying biological processes remains a challenge[9]. In this study, we directly investigated the prospective associations between identified proteins, protein modules and health outcomes using longitudinal data from the UK Biobank. All proteins with a significant causal relationship with loneliness were correlated with the incidence of CVD, T2D, stroke and mortality. For example, a one-unit increase in ADM level was associated with a 58% heightened risk of mortality during the follow-up. Furthermore, on average, 7.5% of the association between loneliness and the risk of four diseases and mortality could be attributed to the levels of ADM in plasma based on the Cox-based approach. The mediating effects of ADM on the relationship between loneliness and all five health outcomes were confirmed through mediation analyses based on counterfactuals. Additionally, other MR-identified proteins also showed significant mediating effects for CVD, stroke and mortality. This is the first study to explore the link between loneliness and health outcomes through plasma proteins in humans within a survival context.

However, it is important to note that the lack of an association between plasma proteins, social isolation and loneliness in MR analyses does not necessarily preclude a mechanistic role for candidate proteins in the connection between social relationships and health. In fact, over half of the proteins and protein modules exhibiting a significant phenotypic correlation with social isolation or loneliness demonstrated a noteworthy association with CVD, T2D, stroke and mortality. Remarkably, GDF15 emerged as the top-risk protein for CVD, stroke, dementia, depression and mortality. A recent study identified GDF15 plasma levels as the top robust predictor across 14 categories of disorders and all-cause mortality[67]. However, GDF15 might also exert anti-inflammatory effects by inhibiting macrophage activation, thus potentially playing a protective role[68]. The disparate roles of GDF15 might be mediated by different mechanisms that are not fully understood. Additionally, consistent with our results, GDF15 was not found to be causally associated with diseases such as hypertension, diabetes, heart disease, stroke and cancer[69]. Therefore, the increases in GDF15 observed in pathologies might be a consequence of these diseases, potentially resulting from inflammation caused by these diseases, rather than a cause[70]. It is also possible that associations with disease development are potentially compensatory[71]. These pieces of evidence may imply that GDF15 is not specific to a particular disease, yet this does not rule out its potential mechanistic relevance to neuroinflammation or other processes pertinent to the link between social relationships and health.

The following considerations should be kept in mind when interpreting our findings. First, the measurements of loneliness and social isolation within the UK Biobank do not differentiate between acute and chronic conditions. While acute social isolation/loneliness, activating the SNS and HPA axis, may be adaptive in the short term, persistent activation of these systems is associated with increased inflammatory activity and implicated in the progression of multiple disease states and psychopathologies[72]. Future research should aim to distinguish

between these conditions. Second, we observed a notable decrease in the associations between plasma proteins and measures of social relationships, particularly with loneliness, when accounting for baseline depressive symptoms. Previous research indicated that 75% of the connection between loneliness and dementia[35], as well as 66% of the link between loneliness and mortality[73], could be attributed to depressive symptoms. Additionally, there is a high genetic correlation between loneliness and depression ($r = 0.63$), nearly equivalent the genetic correlation between loneliness and social isolation ($r = 0.65$)[74], indicating a significant co-occurrence of these phenotypes. Third, although the Olink proteomics platform utilized in this study offers a comprehensive measurement of the human proteome, not all circulating proteins were measured. Additionally, we have characterized plasma proteomic signatures of social isolation and loneliness, but we do not know the relationship with protein levels in other tissues as protein levels are known to differ between cell types[75]. Fourth, consistent causal effects of loneliness on the five proteins were inferred using different methods except for CAUSE, indicating the possibility of correlated pleiotropy. Correlated pleiotropy is common and may frequently contribute to false positives in MR studies[76]. Further investigations in larger, more powerful datasets will be needed to determine whether the observed associations are due to causality rather than pleiotropy. Last, an important limitation of this study is the lack of external validation, given that social isolation and loneliness are not typically measured variables. Nevertheless, cross-validation in the UK Biobank and sensitivity analyses supported the robustness of our findings. Future validation in an independent dataset is essential.

In conclusion, this is the first study delineating robust and comprehensive plasma proteomic signatures associated with social isolation and loneliness. The plasma proteome can help bridge the link between social relationships and morbidity and mortality. Comprehending the biology underlying the impact of social relationships on health, particularly the peripheral changes preceding disease, may provide new opportunities for targeted prevention and for effective intervention.

## Methods

### Study cohort
The UK Biobank is a population-based cohort that involves over 500,000 individuals aged 40–69 years old recruited from 22 centres across the United Kingdom between 2006 and 2010[77]. Participant data include genome-wide genotyping, magnetic resonance imaging, electronic health record linkage, blood and urine biomarkers, as well as various other phenotypic endpoints. All participants in the UK Biobank provided informed consent, and ethical approval was obtained from the National Information Governance Board for Health and Social Care and the North West Multi-Centre Research Ethics Committee (ref. 11/NW/0382).

### Defining social isolation and loneliness
Social isolation and loneliness were calculated from scales that have been widely used in previous studies[35,73]. Social isolation was assessed through three dichotomous questions (living alone (1 = yes), social contact (1 = less than monthly) and participation in social activities (1 = less than weekly)), with an individual defined as socially isolated if scoring 2 or 3. Loneliness was evaluated using two binary items akin to questions in the revised University of California, Los Angeles loneliness scale[78]: often feeling lonely (1 = yes) and frequency of confiding in close people (1 = less than once every few months). An individual was classified as lonely if scoring 2.

### Proteomic measurements
The UKB-PPP consortium conducted proteomic profiling on blood plasma samples collected from 53,026 participants at baseline, using the Olink Explore 3072 Proximity Extension Assay between April 2021 and February 2022. The subcohort comprised a randomly selected subset of 45,507 participants at baseline, 6,229 individuals preselected by the UKB-PPP consortium and 1,290 individuals participating in the COVID-19 repeat-imaging study at multiple visits. Consortium-selected and COVID-19 imaging participants showed widespread differences such as demographics and biochemistry markers, reflecting their non-random sampling, whereas the randomly selected baseline participants remained highly representative of the overall UK Biobank[79]. Following stringent quality control (QC) procedures (refer to biobank.ndph.ox.ac.uk/ukb/ukb/docs/PPP_Phase_1_QC_dataset_companion_doc.pdf for detailed information), 2,923 unique proteins were measured across eight panels containing cardiometabolic (I and II), inflammation (I and II), neurology (I and II) and oncology (I and II). The inter- and intraplate coefficients of variation for all Olink panels were below 20% and 10%, respectively. Due to the QC process, different participants had missing values for different proteins, resulting in varying sample sizes for individual proteins. After excluding proteins with missing data exceeding 50% (GLIPR1, NPM1 and PCOLCE), a total of 2,920 unique proteins were included. We utilized the reported normalized protein expression values, and normalized protein expression underwent inverse-rank normalization before analysis[79].

### Genotyping and QC
Genotype imputation data were available for 487,409 participants (version 3, released in May 2017). Two closely related arrays, the Applied Biosystems UK BiLEVE Axiom by Affymetrix (now part of Thermo Fisher Scientific) and the Applied Biosystems UK Biobank Axiom Array, were employed to genotype over 800,000 markers with good genome-wide coverage[80]. We removed samples that had mismatch between genetically inferred and self-reported sex, high genotype missingness or extreme heterozygosity, and those with more than ten putative third-degree relatives. Our analysis was restricted to subjects identified as Caucasians through principal component analysis. We further filtered out SNPs with call rates <95%, minor allele frequency <0.5% and deviation from the Hardy–Weinberg equilibrium with $P < 1 \times 10^{-6}$, using PLINK 2 (ref. 81). After QC procedures, we retained a total of 9,910,057 SNPs and 337,138 participants.

### Structural magnetic resonance imaging data
All neuroimaging data were acquired, preprocessed, quality controlled and made available by the UK Biobank (https://biobank.ctsu.ox.ac.uk/crystal/crystal/docs/brain_mri.pdf). Neuroimaging data were collected with a standard Siemens Skyra 3T scanner with a 32-channel head coil. T1-weighted images were processed with FreeSurfer v6 to derive measurements of cortical and subcortical volumes. The Qoala-T approach was used to check FreeSurfer outputs, with additional manual checks performed for outputs close to the threshold. Any outputs that did not pass QC were excluded[82,83]. A total of 68 cortical regions parcellated by the Desikan–Killiany atlas[84] and 16 subcortical regions defined by the aseg atlas[85] were used in the current study.

### Blood biomarkers
The UK Biobank blood sample collection was undertaken at baseline. Blood biochemistry data (category 17518) were derived from approximately 480,000 participants. Detailed QC procedures can be found at https://biobank.ndph.ox.ac.uk/showcase/ukb/docs/biomarker_issues.pdf. We categorized 30 biochemical markers into five groups, covering liver, renal, hormone, bone and joint, and immunometabolism. Meanwhile, blood count data (category 100081), encompassing 31 haematological markers related to red blood cell, white blood cell and platelet, were collected from the same number of participants. In-depth information about the haematology analysis is available at https://biobank.ndph.ox.ac.uk/showcase/ukb/docs/haematology.pdf. Metabolic biomarkers were measured in randomly selected EDTA plasma samples from approximately 280,000 UK Biobank participants, using a high-throughput NMR-based metabolic biomarker profiling

platform developed by Nightingale Health Ltd. Further details on the processing and QC are described previously[86]. In this study, we utilized absolute concentrations of 168 biomarkers, which were categorized into 16 groups such as amino acids, apolipoproteins and cholesterol. The R package 'ukbnmr'[87] was employed to remove technical variation in NMR data. All 229 blood biomarkers were inverse-rank normalized before analyses.

### Ascertainment of health outcomes

The identification of five diseases (CVD: International Classification of Diseases tenth codes of I20, I21, I22, I23, I24 and I25; all-cause dementia: F00, F01, F02, F03 and G30; T2D: E11; depression: F32 and F33; and stroke: I60, I61, I63 and I64) relied on the 'first occurrence' data fields generated by the UK Biobank ([https://biobank.ndph.ox.ac.uk/showcase/label.cgi?id=1712](https://biobank.ndph.ox.ac.uk/showcase/label.cgi?id=1712)). These data were ascertained through a combination of primary care, hospital in-patient, death register and self-reported information. The date of diagnosis was set as the earliest date of diagnosis, regardless of the source used. Prevalent cases were defined as diagnoses within the first 3 years of follow-up or self-reported cases at baseline, and these cases were excluded to mitigate potential reverse-causation bias. Follow-up for deaths started at inclusion in the UK Biobank study (from national death registers) and ended on 30 November 2022 or upon death, for all participants.

### Covariates

Covariates were selected based on literature and data availability. Demographic factors included age, sex, assessment centre, ethnicity (white, mixed, Asian, Black and other), BMI, household income (<£31,000 or ≥£31000) and education level. Education level was grouped into four categories reflecting similar years of education (college/university degree, education to age 18 years or above, education to age 16 years qualifications and no qualifications)[88]. Current smoking status and alcohol intake frequency (<3 or ≥3 times per week) were obtained by self-rated questions. Depressive symptoms over the past 2 weeks were assessed using four items from the Patient Health Questionnaire[89] (depressed mood, uninterest/absence of enthusiasm, tenseness/restlessness and tiredness/lethargy). Physical activity was assessed using the International Physical Activity Questionnaire short form[90], which includes six questions on the frequency and duration of walking, moderate-intensity exercise and vigorous exercise. Following the scoring protocol[91], responses of 'unable to walk' were recoded to 0, while 'do not know' and 'prefer not to answer' were treated as missing data. Bouts of activity lasting less than 10 min duration were not counted and activity bouts of longer than 4 h were truncated. Weekly minutes of each activity category were multiplied by the corresponding metabolic equivalent (MET) values (that is, 3.3 for walking, 4.0 for moderate physical activity and 8.0 for vigorous physical activity)[92], then summed to obtain the total MET minutes representing the energy expended during physical activity. Technical factors for plasma proteins included batch and time gap between blood collection and protein measurement. The first 20 genetic PCs were controlled for to mitigate potential population stratification.

### Statistical analysis

**PWAS.** Logistic regression was performed with social isolation or loneliness as the outcome and protein level as the predictor. Initially, a simple model was employed, controlling for age, sex, site, technical factors and the first 20 genetic PCs. Subsequently, a fully adjusted model was examined, additionally controlling for ethnicity, education level, household income, smoking, alcohol consumption and BMI. The fully adjusted model was used for all primary analyses.

Several sensitivity analyses were conducted to assess the robustness of these findings: (1) To address the limitations of dichotomous variables potentially losing statistical power, raw scores were used and modelled with ordered logistic regression, controlling for all covariates in the primary model. (2) To address potential bias stemming from the UKB-PPP subcohort composition, the primary analyses were replicated exclusively within the randomly selected subset. (3) PWAS was conducted specifically in Caucasians to further mitigate the possibility of population stratification. (4) Sex and age differences were evaluated by incorporating interaction terms between sex or age and protein levels into logistic models, adjusting for the same covariates as in the primary model. Additionally, subgroup PWAS analyses were conducted separately for males and females, as well as for younger (<60 years) and older (≥60 years) individuals, to provide further insights. (5) Depressive symptoms and physical activity were included to assess their respective impacts on the relationship between social isolation/loneliness and proteins. Social isolation and loneliness were simultaneously included in the model to test for mutual influence. The reduced percentage of OR was calculated to quantify the influence of additional factors. (6) Participants were categorized into four groups: neither isolated nor lonely (SI−LO−), socially isolated but not lonely (SI+LO−), not isolated but lonely (SI−LO+) and socially isolated and lonely (SI+LO+). Multinomial logistic regression was performed with all covariates from the primary model, using the SI−LO− group as the reference. A likelihood ratio test was used to estimate the overall association of protein levels compared with the model with covariates only. Significant proteins related to a specific group required both a significant model fit and a significant coefficient after Bonferroni correction ($P < 0.05/2920 = 1.7 \times 10^{-5}$). (7) Cross-validation was performed by randomly dividing socially isolated, lonely and control participants to ensure similar sample sizes for each group in the two split datasets. Fully adjusted PWAS analyses were performed separately on the two datasets, and this process was repeated 100 times to ensure reliability.

**Protein co-expression network analysis.** Participants with missing protein data exceeding 50% were excluded, and the remaining missing proteins were imputed using the *k*-nearest imputation function in the R package 'impute'. Employing the full set of 2,920 proteins from 46,850 participants, we utilized Netboost[93], a dimension-reduction procedure that extends the widely used weighted gene co-expression network analysis method[94], to identify co-expressed protein networks (modules). The procedure involved three steps: first, we calculated the boosting-based filter and a sparse distance matrix based on Spearman correlation. Then, protein modules were identified through sparse hierarchical clustering and the dynamic tree cut procedure. Finally, we aggregated information in the modules using the first PC. Consistent with prior network-based proteomic analyses[95], we set the minimum module size to 20, applied a Spearman filter method with a soft power of $\beta = 2$ and an 'unsigned' network approach in clustering. Additionally, we compared protein networks constructed with different soft powers (that is, 3 and null) against these parameters. The association between the expression of each protein module and social isolation and loneliness was examined using logistic regression, adjusting for covariates in the fully adjusted model. To assess the consistency between the identified protein networks and the proteins associated with social isolation or loneliness identified through PWAS, two-sided Fisher's exact tests were conducted to determine whether the identified proteins were significantly enriched in the identified networks using all proteins in the module, as well as the top 20% and top 10% of proteins that exhibit the most significant correlation with the corresponding module eigengene.

**PPI network and functional enrichment analysis.** The PPI network of the proteins of interest was constructed using STRING[28] with default settings and visualized by Cytoscape[96]. The cytoHubba plugin[29] in Cytoscape was utilized to identify hub proteins by the MCC method.

To assess the biological relevance of the identified protein sets, gprofiler2 (ref. [97]) was employed to calculate functional enrichments for Gene Ontology (GO)[98] biological processes and molecular functions, Kyoto Encyclopedia of Genes and Genomes (KEGG)[99], Reactome[100] and

WikiPathways[101]. Proteins associated with social isolation/loneliness after FDR correction ($q < 0.05$, corrected for two constructs simultaneously) were used as the test set, with all proteins measured in the UK Biobank ($N = 2,920$) used as the background set. The significance of enrichment was indicated by the $P$ value of the hypergeometric test, followed by FDR correction. We considered results with FDR $q < 0.05$ as statistically significant.

**MR.** We applied bidirectional MR to investigate the causal relationship between identified proteins and protein networks with social isolation and loneliness. The GWAS on plasma protein abundance, using protein level for individual proteins and module eigengene for protein networks, was conducted in Caucasian participants with quality-controlled genotyping data from the UK Biobank. To avoid bias due to participant overlap, separate GWASs for social isolation ($N = 297,396$) and loneliness ($N = 288,696$) were performed using a distinct set of Caucasian participants, excluding those involved in the protein-related GWAS. Age, sex and the first 20 genetic PCs were adjusted for in all GWAS analyses, while protein technical factors were additionally accounted for in GWAS analyses of proteins and protein modules. GWAS analyses were performed through PLINK 2 (ref. 81) and GCTA[102].

For the forward direction (protein to social isolation/loneliness), we selected pQTLs with genome-wide significance ($P < 5 \times 10^{-8}$) as IV and applied clumping (1,000 kb distance, maximum linkage disequilibrium (LD) $r^2$ of 0.001). pQTLs were categorized into *cis*-pQTLs (±1 MB window of the gene encoding the target protein) and *trans*-pQTLs (outside the ±1 MB window). Two types of MR analyses were conducted using *cis*-pQTLs alone and both *cis*- and *trans*-pQTLs. For the backward direction (social isolation/loneliness to protein), due to a limited number of SNPs reaching genome-wide significance after clumping, we relaxed the $P$ value threshold to $1 \times 10^{-6}$ to extract genetic instruments for social isolation and loneliness. We utilized IVW as the primary analytic approach, with the Wald ratio test employed if only one instrument was available. To account for pleiotropy, three sensitivity analyses were conducted: (1) weighted median[103]; (2) MR–Egger[104], which adds an intercept to the IVW regression to exclude confounding from uncorrelated pleiotropy; and (3) generalized summary data-based MR (GSMR) with the heterogeneity in independent instrument-outlier method[105] to identify and remove SNPs with evidence for significant uncorrelated pleiotropy ($P < 0.01$). For GSMR, we used 10,000 randomly selected unrelated samples from the UK Biobank as a reference to determine LD patterns. For significant MR results, CAUSE[76] was performed with a large set of LD-pruned SNPs ($r^2 < 0.01$ with an arbitrary $P \leq 1 \times 10^{-3}$) to further consider correlated pleiotropy. Additionally, Cochran's $Q$-test[106] was used to determine IV heterogeneity. A leave-one-out analysis was performed to assess the influence of individual SNPs on MR estimation. FDR correction was applied to the results of each direction and for social isolation and loneliness separately, with FDR $q < 0.025$ considered significant. MR analyses were performed utilizing the R packages 'TwoSampleMR'[107], 'gsmr2'[105] and 'cause'[76].

**Colocalization analysis.** The results that survived the multiple testing threshold on MR analysis were evaluated using a colocalization analysis to estimate the posterior probability (PP) of each genomic locus containing a single variant affecting both the protein and the phenotype. The analysis was based on a Bayesian model that assesses the probability of a shared causal variant (PPH4) or distinct causal variants (PPH3)[108]. Variants within a ±500 kb window around the significant SNP ($P < 5 \times 10^{-8}$) with the smallest $P$ value were included. Colocalization was performed using with default priors (PP of initial trait association (p1) is $1 \times 10^{-4}$, PP of second trait association (p2) is $1 \times 10^{-4}$ and prior probability of shared causal variant across two traits (p12) is $1 \times 10^{-5}$). Two signals were considered to have a strong support of colocalization if the PPH4 ≥ 0.8. Medium colocalization indication was defined as 0.5 < PPH4 < 0.8. The analysis was performed using the R package 'coloc'[108].

**Associations to other blood biomarkers and brain volumes.** Linear regression was used to investigate the associations between proteins and protein modules related to social isolation/loneliness and various blood biochemical, haematological and metabolic markers, as well as cortical and subcortical volumes. The same covariates as in the primary model were controlled for. Additionally, intracranial volume and the time gap between baseline and imaging collection were considered in neuroimaging analyses.

**Cox proportional hazards model.** Associations between the abundance of social isolation or loneliness associated proteins and five diseases (CVD, dementia, T2D, depression and stroke) and mortality were investigated using Cox proportional hazards models. The proportional hazard assumptions were checked using Schoenfeld residuals and no major violations were observed. All models were adjusted for age, sex, ethnicity, education level, household income, smoking, alcohol consumption, BMI, time gap between blood collection and protein measurement, and the first 20 genetic PCs.

**Mediation analysis.** To support the reliability of our results, we used two approaches to estimate the contribution of proteins explaining the relationship between social isolation/loneliness and health outcomes. First, we estimated the PERM using two Cox proportional hazards models, with the formula $(HR_{(age, sex, site, ethnicity, education level, household income, smoking, alcohol consumption and BMI adjusted)} - HR_{(age, sex, site, ethnicity, education level, household income, smoking, alcohol consumption, BMI and protein adjusted)})/(HR_{(age, sex, site, ethnicity, education level, household income, smoking, alcohol consumption and BMI adjusted)} - 1) \times 100$ (refs. 35,36). Given the limitations of the Cox-based mediation analysis in inferring statistical significance and offering a causal interpretation, we subsequently adopted a counterfactual-based mediation analysis[109]. In this study, we employed a straightforward procedure based on marginal structural models that directly parameterize the natural direct and indirect effects of interest[39]. Based on the results of MR analysis, we found a significant causal relationship between loneliness and protein levels. Therefore, loneliness was treated as the exposure and protein level were used as the mediator. In the mediator model, we included all the covariates used in the PWAS (that is, age, sex, site, batch, time gap between blood collection and protein measurement, ethnicity, education level, household income, smoking, alcohol consumption, BMI and the first 20 genetic PCs). In the outcome model, all covariates except protein technical factors and genetic PCs were adjusted for. The Aalen additive hazard model[110] was utilized for the survival outcome. CIs were computed as the estimate of the natural direct or indirect effect plus/minus 1.96 times a robust standard error.

**Multiple comparison correction.** All reported $P$ values in this study are two sided, except for the hypergeometric test used in the enrichment analysis. We provided results following both Bonferroni and FDR corrections. The Bonferroni correction was calculated as 0.05 divided by the total number of tests, encompassing both social isolation and loneliness as predictors. FDR correction was applied to all analyses simultaneously, with significance set at $q < 0.05$. For MR analysis, due to the substantial difference in the number of proteins associated with social isolation and loneliness, separate FDR corrections were applied for each, with significance set at $q < 0.025$.

#### Reporting summary
Further information on research design is available in the Nature Portfolio Reporting Summary linked to this article.

## Data availability
The data used in the present study are available from the UK Biobank (https://www.ukbiobank.ac.uk) with restrictions applied. Data were used under licence and are thus not publicly available. Details regarding registration for data access can be found at http://www.ukbiobank.ac.uk/register-apply/. The data used in this study were accessed from

 

the UK Biobank under the application number 19542. GWAS summary statistics used can be found via the figshare website at https://figshare.com/projects/GWAS_summary_data/224229 (ref. 111). European ancestry reference data from the 1000 Genomes Project can be found via GitHub at https://github.com/getian107/PRScsx?tab=readme-ov-file.

## Code availability

Custom scripts for the analyses have been made available via the following GitHub repository at https://github.com/chunshen617/Proteomics_loneliness.

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

## Acknowledgements

C.S. is supported by grants from the National Natural Sciences Foundation of China (no. 82101617) and the China Postdoctoral Science Foundation (no. 2022M710765). W.C. is supported by grants from the National Natural Sciences Foundation of China (no. 82071997) and the Shanghai Rising-Star Program (no. 21QA1408700). J.F. is supported by National Key R&D Program of China (no. 2018YFC1312904 and no. 2019YFA0709502), the Shanghai Municipal Science and Technology Major Project (no. 2018SHZDZX01), the 111 Project (no. B18015), Shanghai Center for Brain Science and Brain-Inspired Technology and the Zhangjiang Lab. The funders had no role in study design, data collection and analysis, decision to publish or preparation of the manuscript.

## Author contributions

C.S., W.C. and J.F. conceived and designed the study. C.S. performed the data analysis with support from R.Z. All authors contributed to interpretation of the results. C.S. drafted the paper. B.J.S., C.S., J.Y. and W.C. revised the paper. C.S. and R.Z. contributed to the visualization. C.S., W.C. and J.F had full access to the data in the study and took responsibility for the integrity of the data and the accuracy of the data analysis. All authors read and approved the final manuscript.

## Competing interests

The authors declare no competing interests.

## Additional information

**Correspondence and requests for materials** should be addressed to Barbara J. Sahakian, Wei Cheng or Jianfeng Feng.

Wei Cheng
Barbara Sahakian

# Reporting Summary

## Statistics

For all statistical analyses, confirm that the following items are present in the figure legend, table legend, main text, or Methods section.

| n/a | Confirmed | |
|---|---|---|
| ☐ | ☒ | The exact sample size (*n*) for each experimental group/condition, given as a discrete number and unit of measurement |
| ☐ | ☒ | A statement on whether measurements were taken from distinct samples or whether the same sample was measured repeatedly |
| ☐ | ☒ | The statistical test(s) used AND whether they are one- or two-sided *Only common tests should be described solely by name; describe more complex techniques in the Methods section.* |
| ☐ | ☒ | A description of all covariates tested |
| ☐ | ☒ | A description of any assumptions or corrections, such as tests of normality and adjustment for multiple comparisons |
| ☐ | ☒ | A full description of the statistical parameters including central tendency (e.g. means) or other basic estimates (e.g. regression coefficient) AND variation (e.g. standard deviation) or associated estimates of uncertainty (e.g. confidence intervals) |
| ☐ | ☒ | For null hypothesis testing, the test statistic (e.g. *F*, *t*, *r*) with confidence intervals, effect sizes, degrees of freedom and *P* value noted *Give P values as exact values whenever suitable.* |
| ☐ | ☒ | For Bayesian analysis, information on the choice of priors and Markov chain Monte Carlo settings |
| ☒ | ☐ | For hierarchical and complex designs, identification of the appropriate level for tests and full reporting of outcomes |
| ☐ | ☒ | Estimates of effect sizes (e.g. Cohen's *d*, Pearson's *r*), indicating how they were calculated |

*Our web collection on statistics for biologists contains articles on many of the points above.*

## Software and code

Policy information about availability of computer code

Data collection    No software was used.

Data analysis    R version 4.2.0 was primarily used for the analyses in this study. Related R packages include: Protein co-expression analysis was performed using 'Netboost' (v2.4.1) (https://www.bioconductor.org/packages/release/bioc/html/netboost.html); Protein imputation for network analysis was performed using 'impute' (v1.70.0) (https://bioconductor.org/packages/release/bioc/html/impute.html); Functional enrichment was performed using 'gprofiler2' (v0.2.3) (https://cran.r-project.org/web/packages/gprofiler2/index.html); Protein GWASs were performed through GCTA (v1.94.1) (https://yanglab.westlake.edu.cn/software/gcta); Mendelian randomization analysis was performed using 'TwoSampleMR' (v0.5.7) (https://mrcieu.github.io/TwoSampleMR/), 'gsmr2' (v1.1.1) (https://yanglab.westlake.edu.cn/software/gsmr/), and 'cause' (v1.2.0) (https://github.com/jean997/cause); Colocalization was performed through 'coloc' (v5.2.3) (https://chr1swallace.github.io/coloc/); Preprocessing of NMR-metabolic data was performed using 'ukbnmr' (v2.2) (https://cran.r-project.org/web/packages/ukbnmr/index.html). Social isolation and loneliness GWASs were performed through PLINK 2.0 (https://www.cog-genomics.org/plink/2.0/). PPI was performed using STRING (v12.0) (https://string-db.org/cgi/input?sessionId=besj5Fp9GYc2&input_page_show_search=on). PPI network visualization was conducted by Cytoscape (v3.10.0) (https://cytoscape.org/). T1-weighted images were processed with Freesurfer (v6.0.0) (https://surfer.nmr.mgh.harvard.edu/). Custom scripts for the analyses have been made available through the following GitHub repository: https://github.com/chunshen617/Proteomics_loneliness.

For manuscripts utilizing custom algorithms or software that are central to the research but not yet described in published literature, software must be made available to editors and reviewers. We strongly encourage code deposition in a community repository (e.g. GitHub). See the Nature Portfolio guidelines for submitting code & software for further information.

## Data

The data used in the present study are available from the UK Biobank (https://www.ukbiobank.ac.uk) with restrictions applied. Data were used under licence and are thus not publicly available. Details regarding registration for data access can be found at http://www.ukbiobank.ac.uk/register-apply/. The data used in this study were accessed from the UK Biobank under the application number 19542. GWAS summary statistics used can be found at the Figshare website (https://figshare.com/projects/GWAS_summary_data/224229) (ref. 111). European ancestry reference data from the 1000 Genomes Project can be found via https://github.com/getian107/PRScsx?tab=readme-ov-file.

## Research involving human participants, their data, or biological material

| | |
|---|---|
| Reporting on sex and gender | Both male and female subjects from the UK Biobank study were included. Sex (Data-Field 31) was obtained from the central registry at recruitment, but in some instances, it was updated by the participant. Consequently, this field may contain a blend of the sex recorded by the NHS for the participant and self-reported sex. Summary statistics on sex distributions were reported in Table 1. All statistical models were adjusted for sex, and sensitivity analyses including a sex interaction term and sex subgroup analysis were also conducted. |
| Reporting on race, ethnicity, or other socially relevant groupings | Table 1 presents the baseline characteristics of study participants, encompassing age, sex, ethnicity (white, mixed, Asian, black, and other), education level, and household income. Covariates were chosen based on literature and data availability. Two types of models, each with distinct covariates, were investigated in this study. The simple model comprised age, sex, site, technical factors, and the first 20 genetic PCs. Additionally, a fully adjusted model was explored, incorporating ethnicity, education level, household income, smoking, alcohol consumption, and BMI as covariates. The fully adjusted model was utilized for all primary analyses. |
| Population characteristics | Our primary study population included 42,062 participants (56.4±8.2 years and 52.3% female) from the UK Biobank, who had quality-controlled proteomic data and complete behavioral data including social isolation, loneliness, and all covariates. A flow chart of participant selection is shown in Supplementary Fig. 1. Among these, 3,905 (9.3%) reported as being socially isolated, and 2,689 (6.4%) felt lonely. Detailed demographic characteristics, stratified by social isolation and loneliness, are presented in Table 1. During a median (SD) follow-up of 13.7 (2.1) years (ended on November 30, 2022 or death), 2,695 participants developed CVD, 892 developed all-cause dementia, 1,703 developed T2D, 1,521 developed depression, 983 developed stroke, and 4,255 passed away. |
| Recruitment | The UK Biobank is a population-based cohort that involves over 500,000 individuals aged 40–69 years recruited from 22 centers across the UK between 2006 and 2010. Previous investigation have demonstrated a healthy volunteer bias in the UK Biobank. |
| Ethics oversight | All participants in the UK Biobank provided informed consent, and ethical approval was obtained from the National Information Governance Board for Health and Social Care and the North West Multi-Centre Research Ethics Committee (ref: 11/NW/0382). |

Note that full information on the approval of the study protocol must also be provided in the manuscript.

# Field-specific reporting

Please select the one below that is the best fit for your research. If you are not sure, read the appropriate sections before making your selection.

☒ Life sciences          ☐ Behavioural & social sciences          ☐ Ecological, evolutionary & environmental sciences

# Life sciences study design

All studies must disclose on these points even when the disclosure is negative.

| | |
|---|---|
| Sample size | No statistical methods were used to predetermine sample sizes and all currently available sample in the UK Biobank were used. Our primary study population included 42,062 participants (56.4±8.2 years and 52.3% female) from the UK Biobank, who had quality-controlled proteomic data and complete behavioral data including social isolation, loneliness, and all covariates. |
| Data exclusions | In survival analyses, we excluded prevalent cases, which were defined as diagnoses occurring within the first three years of follow-up or self-reported cases at baseline. This exclusion was implemented to reduce the risk of potential reverse-causation bias.<br>Genome-wide association studies (GWASs) on plasma protein abundance were conducted in Caucasian participants with qualified genotyping |

| Replication | data from the UK Biobank. To avoid over-fitting in two-sample MR, GWASs for social isolation (N=297,396) and loneliness (N=288,696) were performed using a distinct set of Caucasian participants, excluding those involved in the protein-related GWASs. |

| Replication | We replicated the primary analyses exclusively within the randomly selected subset (N=36,250). The proteins identified in this subset were consistent with those in all available samples, and the proteomic association patterns in these two populations were highly correlated (Supplementary Fig. 4). Additionally, to mitigate potential population stratification, we conducted PWAS specifically in Caucasians (N=35,697) and observed that the proteomic associative patterns were highly correlated with those found in the full sample (Supplementary Fig. 7). We also performed cross-validation by randomly splitting the UK Biobank samples 100 times. Our results showed that most of the significant proteins identified using the full sample retained significance in at least one of the two split samples, and proteomic associative patterns for social isolation and loneliness between the two split samples exhibited medium to large correlations (Supplementary Fig. 14). |

| Randomization | The primary analyses were controlled for age, sex, site, technical factors, ethnicity, education level, household income, smoking, alcohol consumption, BMI, and the first 20 genetic PCs. Additionally, two types of subgroup analyses were undertaken to examine potential sex (male vs female) and age (<60 vs >=60 years) differences. |

| Blinding | Blinding was not applicable to this study as this study is observational. |

# Reporting for specific materials, systems and methods

We require information from authors about some types of materials, experimental systems and methods used in many studies. Here, indicate whether each material, system or method listed is relevant to your study. If you are not sure if a list item applies to your research, read the appropriate section before selecting a response.

## Materials & experimental systems

| n/a | Involved in the study |
|---|---|
| ☐ | ☒ Antibodies |
| ☒ | ☐ Eukaryotic cell lines |
| ☒ | ☐ Palaeontology and archaeology |
| ☒ | ☐ Animals and other organisms |
| ☒ | ☐ Clinical data |
| ☒ | ☐ Dual use research of concern |
| ☒ | ☐ Plants |

## Methods

| n/a | Involved in the study |
|---|---|
| ☒ | ☐ ChIP-seq |
| ☒ | ☐ Flow cytometry |
| ☐ | ☒ MRI-based neuroimaging |

## Antibodies

| Antibodies used | The UK Biobank plasma samples were analysed using the Olink Explore 3072 proximity extension assay platform, which is based upon an in-solution binding of two polyclonal antibody pools to a target protein and subsequent hybridization and enrichment of two unique single-stranded DNA probes to create a double stranded barcode unique for the antigen. The platform consists of 2,941 immunoassays targeting 2,925 proteins. Each assay is based on a pair of polyclonal antibodies. The antibodies bind to different sites on the target protein and are labelled with single-stranded complementary oligonucleotides. If matching pairs of antibodies bind to the protein, the attached oligonucleotides hybridize, and are then measured using next-generation sequencing. Olink Explore 3072 consists of 8 panels of 384 assays analysed by next-generation sequencing. |

| Validation | Quality control protocol was implemented by the UK Biobank Pharma Proteomics Project, and was developed and approved by scientists across the thirteen participating biopharmaceutical companies, including Amgen, Alnylam, AstraZeneca, Biogen, Bristol Myers Squibb, Calico, Genentech, GlaxoSmithKline, Janssen (Johnson & Johnson), Novo Nordisk, Pfizer, Regeneron, and Takeda. More detailed information can be found in biobank.ndph.ox.ac.uk/ukb/ukb/docs/PPP_Phase_1_QC_dataset_companion_doc.pdf. |

## Plants

| Seed stocks | Report on the source of all seed stocks or other plant material used. If applicable, state the seed stock centre and catalogue number. If plant specimens were collected from the field, describe the collection location, date and sampling procedures. |

| Novel plant genotypes | Describe the methods by which all novel plant genotypes were produced. This includes those generated by transgenic approaches, gene editing, chemical/radiation-based mutagenesis and hybridization. For transgenic lines, describe the transformation method, the number of independent lines analyzed and the generation upon which experiments were performed. For gene-edited lines, describe the editor used, the endogenous sequence targeted for editing, the targeting guide RNA sequence (if applicable) and how the editor was applied. |

| Authentication | Describe any authentication procedures for each seed stock used or novel genotype generated. Describe any experiments used to assess the effect of a mutation and, where applicable, how potential secondary effects (e.g. second site T-DNA insertions, mosiacism, off-target gene editing) were examined. |

# Magnetic resonance imaging

## Experimental design

**Design type**

> Structural MRI

**Design specifications**

> The UK Biobank designed the imaging acquisition protocols including 6 modalities, covering structural, diffusion and functional imaging. The collection order is T1-weighted structural image, resting-state functional MRI, task functional MRI, T2-weighted FLAIR structural image, diffusion MRI and susceptibility-weighted imaging. The T1-weighted structural image was acquired using straight sagittal orientation for 5 minutes.

**Behavioral performance measures**

> We used T1-weighted structural imaging, during which participants were not required to perform any tasks.

## Acquisition

**Imaging type(s)**

> T1-weighted structural imaging

**Field strength**

> 3T

**Sequence & imaging parameters**

> The EPI-based acquisitions utilize simultaneous multi-slice (multiband) acceleration. UK Biobank uses pulse sequences and reconstruction code from the Center for Magnetic Resonance Research (CMRR), University of Minnesota https://www.cmrr.umn.edu/multiband. The resolution is 1x1x1 mm and the field of view is 208x256x256 matrix. Straight sagittal orientation is used. TR and TE are 2000ms and 2.01ms respectively. The flip angle is 8 deg. Detailed sequence and imaging parameters are openly available here: https://biobank.ndph.ox.ac.uk/showcase/showcase/docs/brain_mri.pdf.

**Area of acquisition**

> Whole brain

**Diffusion MRI**  ☐ Used  ☒ Not used

## Preprocessing

**Preprocessing software**

> Imaging-derived phenotypes (IDPs) generated through an image-processing pipeline developed and run on behalf of the UK Biobank were used in this study (https://biobank.ctsu.ox.ac.uk/crystal/crystal/docs/brain_mri.pdf). T1-weighted images were processed using FreeSurfer. Where available, T2_FLAIR images were used in conjunction with T1-weighted images to achieve more accurate cortical modeling than with T1 alone. Surface atlases were employed to extract IDPs related to surface area, volume, and mean cortical thickness of standard atlas regions. The Qoala-T approach was used to assess the quality of FreeSurfer outputs, supported by manual checks for outputs near the quality threshold. Any FreeSurfer outputs failing quality control were excluded from the IDPs. Surface volumes extracted using the Desikan-Killiany atlas for 68 cortical regions and the aseg atlas for 16 subcortical regions were used in this study. The full processing pipeline is openly available here: https://doi.org/10.1016/j.neuroimage.2017.10.034 and https://www.nature.com/articles/nn.4393.

**Normalization**

> Spatial normalization was performed whenever applicable as described in detail in Miller et al. Nature Neuroscience (2016) and Alfaro-Almagro et al. Neuroimage (2018).

**Normalization template**

> 1 mm resolution version of MNI152 template

**Noise and artifact removal**

> Noise and artifact removal were described in detail in Miller et al. Nature Neuroscience (2016) and Alfaro-Almagro et al. Neuroimage (2018).

**Volume censoring**

> No volume censoring performed on this data

## Statistical modeling & inference

**Model type and settings**

> Linear regression model was used to investigate the associations between proteins and IDPs. Covariates include age, sex, imaging collection site, batch, time gap between blood collection and protein measurement, ethnicity, education level, household income, smoking, alcohol consumption, BMI, the first 20 genetic PCs, ICV, and time gap between baseline and imaging collection.

**Effect(s) tested**

> T-tests were used to derive the two-sided p value.

**Specify type of analysis:**  ☐ Whole brain  ☒ ROI-based  ☐ Both

**Anatomical location(s)**

> Surface volumes extracted using the Desikan-Killiany atlas for 68 cortical regions and the aseg atlas for 16 subcortical regions were used in this study.

**Statistic type for inference**

> No whole brain voxel-wised or cluster-based analyses involved in this study.

(See Eklund et al. 2016)

| Correction | Both Bonferroni (P < 0.05/(84*5) ) and FDR correction (q < 0.05) using the Benjamini-Hochberg procedure results are provided. For FDR correction, all tests are considered simultaneously. |
|---|---|

## Models & analysis

| n/a | Involved in the study |
|---|---|
| ☒ ☐ | Functional and/or effective connectivity |
| ☒ ☐ | Graph analysis |
| ☒ ☐ | Multivariate modeling or predictive analysis |

