## [Peer Review File · Nature Human Behaviour]

Plasma proteomic signatures of social isolation and loneliness associated with morbidity and mortality

Corresponding Author: Professor Jianfeng Feng

Version 0:

Decision Letter:

25th March 2024

Dear Prof Feng,

Thank you once again for your manuscript, entitled "Plasma proteomic signatures of social isolation and loneliness: Associations to morbidity and mortality", and for your patience during the peer review process.

Your Article has now been evaluated by 3 referees. You will see from their comments copied below that, although they find your work of potential interest, they have raised quite substantial concerns. In light of these comments, we cannot accept the manuscript for publication, but would be interested in considering a revised version if you are willing and able to fully address reviewer and editorial concerns.

We hope you will find the referees' comments useful as you decide how to proceed. If you wish to submit a substantially revised manuscript, please bear in mind that we will be reluctant to approach the referees again in the absence of major revisions. We are committed to providing a fair and constructive peer-review process. Do not hesitate to contact us if there are specific requests from the reviewers that you believe are technically impossible or unlikely to yield a meaningful outcome.

In particular, along all other comments the reviewers make, we ask you address all of the following points as a priority:

- [1] repeat the PWAS with proper adjustments to address concerns about unaccounted confounding factors, as requested by both Reviewers #1 and #2;
- [2] perform the alternative MR analyses, as requested by Reviewer #2;
- [3] repeat analyses treating loneliness as a continuous variable;
- [4] although the current sample is the largest available, it will be important to replicate your key findings in an independent cohort, as noted by Reviewer #1.

If you wish to submit a suitably revised manuscript, we would hope to receive it within 4 months. I would be grateful if you could contact us as soon as possible if you foresee difficulties with meeting this target resubmission date.

- Include a "Response to the editors and reviewers" document detailing, point-by-point, how you addressed each editor and referee comment. If no action was taken to address a point, you must provide a compelling argument. When formatting this document, please respond to each reviewer comment individually, including the full text of the reviewer comment verbatim followed by your response to the individual point. This response will be used by the editors to evaluate your revision and sent back to the reviewers along with the revised manuscript.
- Highlight all changes made to your manuscript or provide us with a version that tracks changes.

Link Redacted

Thank you for the opportunity to review your work. Please do not hesitate to contact me if you have any questions or would like to discuss the required revisions further.

[redacted]

Reviewer expertise:

Reviewer #1: Proteomics, PWAS, MR

Reviewer #2: Proteomics, PWAS, MR

Reviewer #3: Proteomics, PWAS, social isolation, loneliness

REVIEWER COMMENTS:

Reviewer #1:

Remarks to the Author:

The authors performed an omics /proteomics study to understand the involvement or circulating proteins and links between social isolation, loneliness and their comorbidity with common diseases.

The association, MR and mediation analyses seems solid, well designed and well described in the manuscript and I find the manuscript well-written as well.

While the results are interesting from a social and well being perspective, I would be suspicious of unaccounted confounding factors in PWAS. From the study design point of view the authors included multiple covariates for adjustment, except physical activity which may be accountable for a large part of the associations. Data on physical activity are collected and available in the UKBiobank cohort. Another limitation is that there is no replication of the PWAS results.

Finally I am wondering the suitability of this omics paper in terms of its and heavy genetic epidemiology jargon for the audience of Nature Human Behavior.

Reviewer #2:

Remarks to the Author:

The authors present a multitude of analyses that examine the association between blood proteins, social isolation/loneliness, and consequences to physical health. The sample is well powered, one of the largest, if not the largest datasets that have been assayed for a multitude of proteins, which should provide an unprecedented level of statistical power. The findings, if validated, are intriguing, as they imply some mechanisms for the negative influence of social isolation/loneliness on health. I have several questions about the analyses performed, which bring the validity of findings into question. I offer some alternative approaches to address these limitations:

Limitations of PWAS: In their PWAS, approximately 25% of proteins were significantly associated with social isolation. This is a very strong result, to the point where it seems implausible. Once the authors adjust for several confounding factors, approximately 6% of proteins are significant. While this number is much more plausible, the selection of covariates still brings up the possibility of residual confounding. First, I suspect residual confounding due to population stratification. Specifically, the ethnicity variable analyzed by the authors was simply a binary variable measuring "white/non-white", which is inadequate for addressing the nuances of ancestral differences between samples. I point out that Sun et al. have recently identified plasma proteomic associations with genetics in this sample (<https://doi.org/10.1038/s41586-023-06592-6>). Given that the alleles determining protein abundances have different frequencies across populations, the potential for population stratification influencing the results of this manuscript is high. Because the authors have access to the genotype data, they should calculate and adjust specifically for principal components derived from genotypes (rather than the categorical ethnicity variable). The authors should also consider a sensitivity analyses, where PWAS are performed only in the European-ancestry subset of data (~90% of the overall UKBB sample). Second, as far as other covariates used to address confounding, it is unclear to me why they split variables into groups. e.g. BMI was split into three groups, rather than directly adjusting for variables continuous measures. The approach of using dichotomized/trichotomized variables can result in residual confounding, as discussed by Groenwald et al. (doi: 10.1503/cmaj.120592).

To address these concerns, the authors need to perform PWAS with genetic PCs and the continuous measures of the potential confounders (in the case of categorical variables, they should model all levels of the variable, rather than combining categories).

Limitations of MR: Authors focus primarily on the Wald ratio test, but also use the weighted median and MR Egger regression. The exposure and outcome samples overlap, which means that MR Egger regression is not appropriate for use (see Minelli et al DOI: 10.1093/ije/dyab084). As well, empirically, MR Egger is very under-powered. As an alternative to the approach taken (i.e., the methods implemented by the TwoSampleMR package), I suggest that they use the MR method developed by Zhu et al., which was specifically designed for eQTL data (<https://doi.org/10.1038/ng.3538>).

Authors performed analysis of cis and trans pQTLs. Trans-pQTLs require some additional analytic considerations: MR assumes that instruments influence the outcome only through modifying the exposure variable. Yet, Sun et al. suggest that trans-pQTLs effects influence multiple domains (<https://doi.org/10.1038/s41586-023-06592-6>). This increases the possibility that the results of

trans-pQTL MR were confounded by correlated horizontal pleiotropy. While authors somewhat considered uncorrelated horizontal pleiotropy, they did not consider correlated horizontal pleiotropy, as detailed by Morrison et al (DOI: 10.1038/s41588-020-0631-4). For trans-pQTL analyses, I suggest the authors also attempt to implement CAUSE MR analyses or LHC-MR analyses (<https://doi.org/10.1038/s41467-021-26970-w>).

Protein networks limitation: How sensitive were the protein network results to the selection of parameters in the Netboost analysis? It would be concerning if major changes happened based on minor perturbations of parameter choice, which seemed arbitrary (authors can correct me if I am wrong about this).

Other limitations: Why did authors choose to examine social isolation and loneliness as binary measures? It seems like there would be a loss of statistical power from dichotomizing these phenotypes, rather than assessing them on a continuum. It would be a shame if this variable selection was only due to the statistical complexity involved (i.e. due to available software, that it is easier to model them as binary variables in logistic regression rather than as ordinal ones).

Reviewer #3:

Remarks to the Author:

Shen et al. used a proteomics approach to examine the proteins associated with social isolation and loneliness, the causal roles of social isolation/loneliness on these proteins, and the mediation effects of the proteins between social isolation/loneliness and diseases/mortality. This is a comprehensive, well-executed, and well-written study. It addresses an important area of research – how social risk factors “get under the skin”. I do not have any major concerns about the methods, results, and inference. The comments below are mainly for clarification, but I think addressing these comments will help the paper further. I do think there is an error with the figure numbering in the main text though. It should be corrected in the revision.

Abstract

1. Does the “MR-identified proteins” refer to the 8 proteins or the 2 proteins (ADM and ASGR1). If it is the 8 proteins, I think it would help to list them.

Introduction

1. Line 80 – 82: why did the author investigate the associations of the MR-identified proteins with brain volume specifically? Why not measurement of physical function?

Results

1. It is hard to follow the sample sizes for different analyses. It is stated at the beginning of this section that the study population included 42,204 participants. However, the results for GDF15 and PCSK9 specified $N = 41,534$, which is also different from the sample size for CXCL14. Was the sample size different because different people had missing values for different proteins, e.g., due to quality control? It could help if the authors state the reason.

2. Continuing with the last comment, there are other sample sizes that need clarification. The sample sizes for social isolation and loneliness were different. The numbers under the “Proteomic measurements” section do not add up. 46,595 participants at baseline, 6,376 individuals pre-selected by the UKB-PPP consortium, and 1,268 individuals participating in the COVID-19 repeat-imaging study at multiple visits do not add up to 54,219 participants from whom blood plasma samples were collected. Then, the sensitivity analysis “exclusively within the randomly selected subset ($N=36,467$)” does not match with any of the numbers above. I think it will help if a flow chart is added to the supplement figure connecting all the sample sizes and statement the reasons for exclusion.

3. Line 146-148: state at the end of the sentence the sample size since it is different from the main analysis. I think it is important because sample size could have been a factor why the number of significant proteins reduced.

4. Sensitivity analysis additionally adjusted for depression: I wonder why Supplementary Fig. 7A and 7B did not use the same beta-beta plots as in Supp. Fig. 4 B and D. I cannot reconcile the statement in the main text that the adjustment attenuated the associations and the lack of shift in the z-statistics distributions in Supp. Fig. 7A and 7B. I also think that another volcano plot can help here, in case a different set of proteins were significant after additional adjustment of depression. That’s the overlap of the z-score distribution in A and B was due to associations for some proteins being strengthened. The same comment applies to Supp. Fig. 8.

5. I wonder if the authors have considered creating 4 categories of social isolation and loneliness, i.e., socially isolated but not lonely (these people may just enjoy being alone), not socially isolated but lonely, socially isolated and lonely, and not isolated and not lonely. Would such intersections give more insights to the underlying construct of impoverished social relationships?

6. I think the main figure numbering was off. The main figure starts with Figure 2 in the text but with Figure 1 in the figures.

7. Line 191 – 195: I think this sentence needs some clarification. Did the fisher’s exact test test the overlap between the top 20% of proteins in each module with the significant proteins from the PWAS, or the overlap between all the proteins in a module with the significant proteins from PWAS? If it is the former, why the top 20% not all the proteins in a module?

8. Line 195 – 197: I wonder if the conclusion of consistent finding was solely based on the overlap between the proteins in the module and the proteins from PWAS. I wonder if the directions of association were or should be considered. For example, PCSK 9, a protein associated with elevated risk of loneliness, was part of Module 3, which showed associated with lower risk of loneliness. I recognized that the negative association of Module 3 may be driven by other proteins in the module, but this was not shown in any figures or tables.

9. I am confused about the mediation analysis. Why were the PERM kept in the paper if there are obvious limitations and another method is better. Cannot the PERM be estimated using natural indirect effect / (natural indirect effect + natural direct effect) from the counterfactual-based analysis (which the authors calculated)? If I am not mistaken, the first approach assumes no exposure-mediator interaction whereas the second approach makes no such assumption. It is weird to me that the PERM was estimated with that assumption and the significance was determined without that assumption (Figure 5 based on figure numbering).

Methods

1. It is stated that the sensitivity analysis for sex and age was performed independently in the stratified groups, so I assumed that no interaction terms were used in the statistical model. If that's the case, can you elaborate further how exactly the p-values were obtained, e.g., what test or statistics were used?
2. Two sample MR for Module 3: How was SNPs selected for Module 3? Was it SNPs associated with any proteins in Module 3? It is not described in Methods.
3. Did I miss it or there is no method description for the associations of the proteins and blood biomarkers and brain volumes?
4. The counterfactual-based method (ref 34) requires a model of the exposure given confounders and a model of the probability densities of continuous mediator (proteins and protein modules) given the exposure and confounders. Please describe the specifications of these models. This is important information as the validity of the mediation analysis relies on the correct specification of these models (ref 34).

Discussion

1. Well-written and reasonable. I wonder if authors can discuss the possibility that GDF15 may be dysregulated as a result of underlying disease pathology (hence the strong associations in the Cox models even before clinical manifestation of the diseases) but no causal indication in the MR. GDF15 was not found to be causally associated with diseases including hypertension, diabetes, heart disease, stroke, and cancer. [1] Also, my understanding is that GDF15 has an anti-inflammatory function [2], yet GDF15 was associated with higher risk of all diseases. What do the authors think about this counterintuitive findings?

[1] Tanaka T, Basisty N, Fantoni G, et al. Plasma proteomic biomarker signature of age predicts health and life span. *Elife*. 2020;9:e61073. Published 2020 Nov 19. doi:10.7554/eLife.61073

[2] Pence BD. Growth Differentiation Factor-15 in Immunity and Aging. *Front Aging*. 2022 Feb 9;3:837575. doi: 10.3389/fragi.2022.837575. PMID: 35821815; PMCID: PMC9261309.

Version 1:

Decision Letter:

15th August 2024

Dear Prof Feng,

Thank you once again for your revised manuscript, entitled "Plasma proteomic signatures of social isolation and loneliness: Associations to morbidity and mortality," and for your patience during the re-review process.

Your manuscript has now been evaluated by Reviewers #1 and #3 from the original round of review. All reviewer feedback is included at the end of this letter. Although the reviewers found your manuscript to have improved during revision, Reviewer #3 maintains some important outstanding concerns. We remain interested in the possibility of publishing your study in *Nature Human Behaviour*, but would like to consider your response to these outstanding concerns in the form of a revised manuscript before we make a decision on publication.

In sum, we invite you to revise your manuscript taking into account all reviewer and editor comments. We are committed to providing a fair and constructive peer-review process. Do not hesitate to contact us if there are specific requests from the reviewers that you believe are technically impossible or unlikely to yield a meaningful outcome.

We hope to receive your revised manuscript within 4-8 weeks. I would be grateful if you could contact us as soon as possible if you foresee difficulties with meeting this target resubmission date.

- Include a "Response to the editors and reviewers" document detailing, point-by-point, how you addressed each editor and referee comment. If no action was taken to address a point, you must provide a compelling argument. This response will be used by the editors and reviewers to evaluate your revision.
- Highlight all changes made to your manuscript or provide us with a version that tracks changes.

Link Redacted

We look forward to seeing the revised manuscript and thank you for the opportunity to review your work. Please do not hesitate to contact me if you have any questions or would like to discuss these revisions further.

[redacted]

Reviewer expertise:

Reviewer #1: Proteomics, PWAS, MR

Reviewer #3: Proteomics, PWAS, social isolation, loneliness

REVIEWER COMMENTS:

Reviewer #1:

Remarks to the Author:

The authors performed all additional analysis I suggested and made necessary changes in the manuscript.

Reviewer #3:

Remarks to the Author:

The authors addressed most of my comments, but I have follow-up comments. These two comments should be addressed before the publication of the paper:

1. The authors confirmed that the sensitivity analysis for sex and age was performed independently in the stratified groups and interaction terms with sex or age were not included in the models. But it is still unclear how the p-value in Line 152-154 "Nevertheless, three proteins (fatty acid-binding protein 4 [FABP4], adrenomedullin [ADM], and GDF15; $P < 0.05/(175+26)$) showed a more pronounced association with loneliness in males than in females (Supplementary Fig. 8C and 8D)." The authors' response mentioned that "two-sided p-values were used to test significance" but for which term was the p-value calculated? Why did the author choose not to use the interaction term with sex or age which is more common approach to test effect modification? I could not find the test of the sex and age difference in the codes provided.

2. Based on the authors' response, the models for the counterfactual-based mediation analysis only adjusted for age and sex for both mediator (protein) models and outcome (disease and mortality) models. The codes confirmed this. However, this is inconsistent with the PWAS analysis which adjusted a lot more covariates. If those covariates in the PWAS were considered confounders for loneliness-protein associations, why were they not confounders anymore in the mediation analysis? Going back to my original comments, the counterfactual-based method requires the outcome model and mediator models to be correctly specified. Including only age and sex is unlikely to satisfy this requirement. At least, the authors should include all the covariates used in the PWAS for the protein models and consider whether these covariates are confounder for the outcome models.

Version 2:

Decision Letter:

Our ref: NATHUMBEHAV-24010198B

30th September 2024

Dear Dr Feng,

Thank you for submitting your revised manuscript "Plasma proteomic signatures of social isolation and loneliness: Associations to morbidity and mortality" (NATHUMBEHAV-24010198B). It has now been seen by one of the original referees and their comments are below. As you can see, the reviewer finds that the paper has improved in revision. We will therefore be happy in principle to publish it in Nature Human Behaviour, pending minor revisions to comply with our editorial and formatting guidelines.

We are now performing detailed checks on your paper and will send you a checklist detailing our editorial and formatting requirements within two weeks. Please do not upload the final materials and make any revisions until you receive this additional information from us.

[redacted]

Reviewer #3 (Remarks to the Author):

The authors address both of my comments of the 1st revision. Well-done!

Version 3:

Decision Letter:

Dear Prof Feng,

We are pleased to inform you that your Article "Plasma proteomic signatures of social isolation and loneliness associated with morbidity and mortality", has now been accepted for publication in *Nature Human Behaviour*.

Please note that *Nature Human Behaviour* is a Transformative Journal (TJ). Authors may publish their research with us through the traditional subscription access route or make their paper immediately open access through payment of an article-processing charge (APC). Authors will not be required to make a final decision about access to their article until it has been accepted. [Find out more about Transformative Journals](https://www.springernature.com/gp/open-research/transformative-journals)

We welcome the submission of potential cover material (including a short caption of around 40 words) related to your manuscript; suggestions should be sent to *Nature Human Behaviour* as electronic files (the image should be 300 dpi at 210 x 297 mm in either TIFF or JPEG format). Please note that such pictures should be selected more for their aesthetic appeal than for their scientific content, and that colour images work better than black and white or grayscale images. Please do not try to design a cover with the *Nature Human Behaviour* logo etc., and please do not submit composites of images related to your work. I am sure you will understand that we cannot make any promise as to whether any of your suggestions might be selected for the cover of the journal.

[redacted]

P.S. Click on the following link if you would like to recommend Nature Human Behaviour to your librarian
<http://www.nature.com/subscriptions/recommend.html#forms>

** Visit the Springer Nature Editorial and Publishing website at http://editorial-jobs.springernature.com?utm_source=ejp_NHumB_email&utm_medium=ejp_NHumB_email&utm_campaign=ejp_NHumB for more information about our career opportunities. If you have any questions please click [here](mailto:editorial.publishing.jobs@springernature.com).

Nature Human Behaviour, Manuscript #: NATHUMBEHAV-24010198

“Plasma proteomic signatures of social isolation and loneliness: Associations to morbidity and mortality” by Chun Shen, Ruohan Zhang, Jintai Yu, Barbara J. Sahakian, Wei Cheng, and Jianfeng Feng

Thank you for the very helpful and constructive comments on this paper. We have carefully revised the paper according to the suggestions received and trust that the paper will soon be accepted for publication. Your suggestions made are shown below in *blue and italics*, and our responses in normal font preceded by --. Changes made to the paper for the revision are shown below within “...”, and in **red** font in the revised paper. The scripts used in this study has been updated and can be found at https://github.com/chunshen617/Proteomics_loneliness.

REVIEWER COMMENTS:

Reviewer #1:

Remarks to the Author:

The authors performed an omics /proteomics study to understand the involvement or circulating proteins and links between social isolation, loneliness and their comorbidity with common diseases.

The association, MR and mediation analyses seems solid, well designed and well described in the manuscript and I find the manuscript well-written as well.

Response:

-- Thank you for acknowledging the study design, data analysis, and writing. We sincerely appreciate your positive feedback and the time and attention you have devoted to evaluating our work.

While the results are interesting from a social and well-being perspective, I would be suspicious of unaccounted confounding factors in PWAS. From the study design point of view the authors included multiple covariates for adjustment, except physical activity which may be accountable for a large part of the associations. Data on physical activity are collected and available in the UK Biobank cohort.

Response:

-- Thank you for your insightful comment. We acknowledge the potential influence of unaccounted confounding factors, particularly physical activity (PA), on our findings. Evidence suggests that both social isolation (SI) and plasma proteins are related to physical activity (PA), though the relationship between loneliness (LO) and PA is less consistent (Kobayashi and Steptoe, 2018; Schrepft et al., 2019; Williams et al., 2019). We agree that PA could partly explain the associations among these variables and should be considered in the analysis.

In the UK Biobank, PA was assessed with the International Physical Activity Questionnaire short form, which includes six questions on the frequency and duration of walking, moderate-intensity exercise, and vigorous exercise (Craig et al, 2003). Following the scoring protocol,

we calculated Metabolic equivalent (MET) minutes by multiplying the weekly minutes of each activity category by the corresponding MET values (IPAQ Research Committee, 2005). We then used the total MET minutes to represent the energy expended during PA. However, we observed that 10,320 out of 49,251 participants with proteomic, SI, and LO data lacked total MET minutes, potentially reducing statistical power. Consequently, we conducted a sensitivity analysis by additionally adjusting for total MET minutes rather than including it in the primary model.

After Bonferroni correction ($P < 0.05 / (2920 \times 2)$), we found that 88 out of 175 significant proteins associated with SI in the primary model remained significant, and 7 out of 26 significant proteins associated with LO remained significant. Following FDR correction applied to all analyses simultaneously, 173 out of 175 significant proteins associated with SI in the primary model retained significance, and all associated with LO remained significant. The overall decrease in significance was due to the reduced sample size (nearly 21% fewer samples). Moreover, the proteomic associative patterns with and without adjustment for physical activity exhibited strong correlations ($r = 0.96$ for SI, and $r = 0.93$ for LO). Thus, the associations between social isolation, loneliness, and plasma proteins were largely independent of physical activity levels.

We have reported the sensitivity analysis results in the Results section and the Supplementary Fig. 11, and introduced the PA measurement in the Methods section.

Results

“Additionally, we examined the potential influence of physical activity (N=34,548) and found that the associations between social isolation, loneliness, and plasma proteins were largely independent of physical activity levels (Supplementary Fig. 11).” (Lines 165-168, Page 5)

Supplementary Fig. 11 PWAS of social isolation and loneliness accounting for physical activity (N=34,548).

A. Volcano plot displays the ORs (x axis) and $-\log_{10}(P)$ -values (y axis) for the association of protein abundance with social isolation. **B.** Scatter plot depicting the z-statistics of the association between social isolation and proteins in the primary model (x axis) alongside the z-statistics of the association between social isolation and proteins further controlling for total MET minutes (y axis). **C.** Volcano plot displays the ORs (x axis) and $-\log_{10}(P)$ -values (y axis) for the association of protein abundance with loneliness. **D.** Scatter plot depicting the z-statistics of the association between loneliness and proteins in the primary model (x axis) alongside the z-statistics of the association between loneliness and proteins further controlling for total MET minutes (y axis).

Methods

“Physical activity was assessed using the International Physical Activity Questionnaire short form (Craig et al, 2003), which includes six questions on the frequency and duration of walking, moderate-intensity exercise, and vigorous exercise. Following the scoring protocol (IPAQ Research Committee, 2005), responses of 'unable to walk' were recoded to 0, while 'do not know' and 'prefer not to answer' were treated as missing data. Bouts of activity lasting less than 10 minutes duration were not counted, and activity bouts of longer than 4 hours were truncated. Weekly minutes of each activity category were multiplied by the corresponding MET values (i.e., 3.3 for walking, 4.0 for moderate physical activity, and 8.0 for vigorous physical activity) (Ainsworth et al, 2011), then summed to obtain the total MET minutes representing the energy expended during physical activity.” (Lines 556-564, Page 14)

Reference

- Kobayashi, L. C., & Steptoe, A. (2018). Social isolation, loneliness, and health behaviors at older ages: longitudinal cohort study. *Annals of Behavioral Medicine*, 52(7), 582-593.
- Schrepft, S., Jackowska, M., Hamer, M., & Steptoe, A. (2019). Associations between social isolation, loneliness, and objective physical activity in older men and women. *BMC public health*, 19, 1-10.

- Williams, S. A., Kivimaki, M., Langenberg, C., Hingorani, A. D., Casas, J. P., Bouchard, C., ... & Wareham, N. J. (2019). Plasma protein patterns as comprehensive indicators of health. *Nature medicine*, 25(12), 1851-1857.
- Craig, C. L., Marshall, A. L., Sjöström, M., Bauman, A. E., Booth, M. L., Ainsworth, B. E., ... & Oja, P. (2003). International physical activity questionnaire: 12-country reliability and validity. *Medicine & science in sports & exercise*, 35(8), 1381-1395.
- IPAQ Research Committee. (2005). Guidelines for data processing and analysis of the International Physical Activity Questionnaire (IPAQ)-short and long forms. <http://www.ipaq.ki.se/scoring.pdf>.
- Ainsworth, B. E., Haskell, W. L., Herrmann, S. D., Meckes, N., Bassett Jr, D. R., Tudor-Locke, C., ... & Leon, A. S. (2011). 2011 Compendium of Physical Activities: a second update of codes and MET values. *Medicine & science in sports & exercise*, 43(8), 1575-1581.

Another limitation is that there is no replication of the PWAS results.

Response:

-- Thank you for raising this important issue. Given that SI and LO are not typically measured variables, it is challenging to find an independent proteomic dataset for validation.

To address this, we conducted a rigorous cross-validation by randomly splitting the UK Biobank samples 100 times. We specifically divided the SI, LO, and control groups in half to ensure similar sample sizes for each group in the two split datasets. Our results demonstrated that a substantial proportion of the significant proteins identified using the full sample retained significance in at least one of the two split samples. For social isolation, 34.3% to 73.7% of the 175 proteins retained significance after Bonferroni correction, with all remaining significant after FDR correction. Notably, all top 10 proteins strongly related to social isolation retained significance after Bonferroni correction. For loneliness, 7.7% to 69.2% of the 26 proteins retained significance after Bonferroni correction, and 53.8% to 100% were significant after FDR correction. Additionally, 9% to 100% of the top 10 proteins strongly related to loneliness retained significance after Bonferroni correction, and 85% to 100% retained significance after FDR correction. Specifically, PCSK9, the protein most strongly related to loneliness, replicated 100% after Bonferroni correction, while ADM, identified as causally related to loneliness through MR and colocalization analysis, survived 98% times after FDR correction. Moreover, we observed medium to large correlations of proteomic associative patterns for social isolation ($r=0.56$ [0.05]) and loneliness ($r=0.35$ [0.08]) between the two split samples. Our comprehensive cross-validation approach demonstrates the robustness and replicability of our results within the constraints of available data.

Moreover, extensive sensitivity analyses, including PWAS within the randomized baseline subset and among Caucasian participants, sex and age subgroup analysis, and accounting for additional covariates such as depressive symptoms and physical activity, indicated that our PWAS discoveries are robust.

Effect sizes seen in the social sciences are often very small (Rosnow & Rosenthal, 2003), necessitating a large sample size to explore such important and complex issues. While these findings require external validation, our study utilized one of the largest and most well-powered

datasets, providing an unprecedented level of statistical power for discovering potential mechanisms underlying the connection between social relationships and health (as noted by Reviewer #2).

We have provided detailed cross-validation results in the sensitivity analysis section and Supplementary Fig. 14, and introduced this analysis in the Methods section. The limitation of lacking external validation has also been acknowledged in the Discussion section.

Results

“Finally, we performed cross-validation by randomly splitting the UK Biobank samples 100 times. Our results showed that most of the significant proteins identified using the full sample retained significance in at least one of the two split samples, and proteomic associative patterns for social isolation ($r=0.56$ [0.05]) and loneliness ($r=0.35$ [0.08]) between the two split samples exhibited medium to large correlations (Supplementary Fig. 14).” (Lines 179-183, Pages 5-6)

Supplementary Fig. 14 Results of cross-validation. **A.** The percentage of replication for all 175 proteins significantly related to social isolation and the top 10 proteins most strongly related to social isolation across 100 iterations. **B.** The distribution of the correlation coefficients for proteomic associative patterns related to social isolation identified in the two randomly split samples. **C.** The percentage of replication for all 26 proteins significantly related to loneliness and the top 10 proteins most strongly related to loneliness across 100 iterations. **D.** The distribution of the correlation coefficients for proteomic associative patterns related to loneliness identified in the two randomly split samples.

Methods

“(7) Cross-validation was performed by randomly dividing socially isolated, lonely, and control participants to ensure similar sample sizes for each group in the two split datasets. Full adjusted PWAS analyses were performed separately on the two datasets, and this process was repeated 100 times to ensure reliability.” (Lines 596-599, Page 15)

Discussion

“Lastly, an important limitation of this study is the lack of external validation, given that social isolation and loneliness are not typically measured variables. Nevertheless, **cross-validation in the UK Biobank and sensitivity analyses supported the robustness of our findings**. Future validation in an independent dataset is essential.” (Lines 441-444, Page 11)

Reference

Rosnow, R. L., & Rosenthal, R. (2003). Effect sizes for experimenting psychologists. *Canadian Journal of Experimental Psychology/Revue canadienne de psychologie expérimentale*, 57(3), 221.

Finally, I am wondering the suitability of this omics paper in terms of its and heavy genetic epidemiology jargon for the audience of Nature Human Behaviour.

Response:

-- This study characterized the proteomic signatures of social isolation and loneliness and investigated the biological pathways through which social relationships influence morbidity and mortality. As highlighted by Reviewer #3, “This is a comprehensive and well-executed, and well-written study. It addresses an important area of research - how social risk factors ‘get under the skin’”. This research aligns closely with the aims and scope of *Nature Human Behaviour*, which seeks to publish research of outstanding significance into any aspect of individual or collective human behaviour, drawing from a broad spectrum of social, biological, health, and physical science disciplines. Therefore, we believe this study will capture the attention and interest of the audience of *Nature Human Behaviour*.

We understand your concern that some background audiences might find the omics and genetic analyses overly complex. To address this, we have clearly explained the purpose of the analyses and the rationale behind them in the Results section. This ensures accessibility and clarity, making the article more approachable for a diverse readership.

Results

“To explore potential interactions among proteins associated with social isolation or loneliness, we analyzed the protein-protein interaction (PPI) networks within the identified pool of 179 proteins using the STRING database.” (Lines 186-188, Page 6)

“Given that complex diseases are not caused by individual proteins but rather result from highly interactive protein networks, we applied a data-driven approach to classify plasma proteins into clusters or modules based on protein co-expression patterns.” (Lines 201-203, Page 6)

“To infer causality between social isolation, loneliness, and the identified proteins and protein modules, we implemented bidirectional two-sample MR.” (Lines 228-229, Page 7)

“For the MR-identified proteins in relation to loneliness, we implemented colocalization analysis to ensure that the results were not confounded by linkage disequilibrium.” (Lines 243-244, Page 7)

“We then extended our analysis to explore the broad associations of the MR-identified proteins with diverse blood biomarkers and brain volume.” (Lines 251-252, Page 7)

“Finally, we delved into the potential mediating role of proteins, which have been implicated as causally linked to loneliness, in the relationship between loneliness and health outcomes.” (Lines 278-279, Page 8)

“Although this approach provided a quantifiable contribution of proteins and is widely used in epidemiological research, the statistical significance and causal interpretation were unclear. Consequently, we used a counterfactual-based mediation analysis to assess the direct and indirect effects in the relationship between loneliness and health outcomes, through the MR-identified proteins.” (Lines 285-289, Page 8)

Reviewer #2:

Remarks to the Author:

The authors present a multitude of analyses that examine the association between blood proteins, social isolation/loneliness, and consequences to physical health. The sample is well powered, one of the largest, if not the largest datasets that have been assayed for a multitude of proteins, which should provide an unprecedented level of statistical power. The findings, if validated, are intriguing, as they imply some mechanisms for the negative influence of social isolation/loneliness on health. I have several questions about the analyses performed, which bring the validity of findings into question. I offer some alternative approaches to address these limitations:

Response:

-- Thank you for endorsing our effort to use the UK Biobank for examining the mechanisms underlying the adverse effects of social isolation/loneliness on health. Additionally, we are grateful for your insightful comments and suggestions regarding statistical analysis. Below, we provide a comprehensive point-by-point response to all of the comments.

Limitations of PWAS: In their PWAS, approximately 25% of proteins were significantly associated with social isolation. This is a very strong result, to the point where it seems implausible. Once the authors adjust for several confounding factors, approximately 6% of proteins are significant. While this number is much more plausible, the selection of covariates still brings up the possibility of residual confounding. First, I suspect residual confounding due to population stratification. Specifically, the ethnicity variable analyzed by the authors was

simply a binary variable measuring “white/non-white”, which is inadequate for addressing the nuances of ancestral differences between samples. I point out that Sun et al. have recently identified plasma proteomic associations with genetics in this sample (<https://doi.org/10.1038/s41586-023-06592-6>). Given that the alleles determining protein abundances have different frequencies across populations, the potential for population stratification influencing the results of this manuscript is high. Because the authors have access to the genotype data, they should calculate and adjust specifically for principal components derived from genotypes (rather than the categorical ethnicity variable). The authors should also consider a sensitivity analyses, where PWAS are performed only in the European-ancestry subset of data (~90% of the overall UKBB sample). Second, as far as other covariates used to address confounding, it is unclear to me why they split variables into groups. e.g. BMI was split into three groups, rather than directly adjusting for variables continuous measures. The approach of using dichotomized/trichotomized variables can result in residual confounding, as discussed by Groenwald et al. (doi: 10.1503/cmaj.120592).

To address these concerns, the authors need to perform PWAS with genetic PCs and the continuous measures of the potential confounders (in the case of categorical variables, they should model all levels of the variable, rather than combining categories).

Response:

-- Thank you for pointing out the possibility of residual confounding due to the selection of covariates. To mitigate potential population stratification, we have categorized ethnicity into five groups: white, mixed, Asian, black, and other (detailed demographic characteristics are in Table 1). Additionally, the first 20 genetic principal components (PCs) were included in both the simple and primary models. Our PWAS results remained consistent before and after these adjustments, with high correlation in proteomic associative patterns (for SI: simple model, $r=0.998$; primary model, $r=0.999$; for LO: simple model, $r=0.992$; primary model, $r=0.985$). The influence on identified proteins was minimal. After Bonferroni correction, 165 out of 175 proteins associated with SI identified previously remained significant, and 26 out of 43 proteins associated with LO remained significant. Importantly, all proteins identified in the original manuscript retained significance in the revised manuscript after FDR correction.

Moreover, we conducted sensitivity analyses by performing PWAS exclusively in Caucasians identified by the genotypes. The proteomic associative patterns in this subgroup were highly correlated with those in the full sample ($r=0.98$ for SI and $r=0.92$ for LO). Specifically, after Bonferroni correction, 80% of the proteins associated with SI identified in the full sample remained significant in Caucasians, and 54% of the proteins associated with LO remained significant. All these proteins remained significant after FDR correction. These results align with findings reported by Sun et al. (2023), which demonstrated no major global enrichment of total pQTLs for any ancestry, with effect sizes being highly consistent between European and non-European ancestries. Collectively, these results strongly suggest that the potential for population stratification influencing our findings is minimal.

Second, following your suggestion, we performed PWAS with continuous BMI as a covariate. The proteomic associative patterns before and after these adjustments showed high correlation, and the impact on the identified proteins was minimal.

We have performed PWAS and other analyses using the revised covariates, updated all the results and figures, and stated the covariates in the Methods section. We have reported PWAS results in Caucasians in the Results section and Supplementary Fig. 7, and introduced this sensitivity analysis in the Methods section.

Figure. Comparison of results before and after revision of covariates. The revised covariates include age, sex, site (22 categories), batch (8 categories), time gap between blood collection and protein measurement (continuous), ethnicity (5 categories), education level (4 categories), household income (2 categories), smoking (2 categories), alcohol consumption (2 categories), BMI (continuous), and the first 20 genetic PCs (continuous). **A.** PWAS results of the primary model for social isolation in the original manuscript. **B.** PWAS results of the primary model for social isolation in the revised manuscript. **C.** PWAS results of the primary model for loneliness in the original manuscript. **D.** PWAS results of the primary model for loneliness in the revised manuscript.

Figure 1 Proteome-wide association study for social isolation and loneliness. **A.** Volcano plot displays the ORs (x-axis) and $-\log_{10}(P\text{-values})$ (y-axis) for the correlation of protein abundance with social isolation. **B.** Volcano plot displays the ORs (x-axis) and $-\log_{10}(P\text{-values})$ (y-axis) for the correlation of protein abundance with loneliness. Covariates included age, sex, site, batch, time gap between blood collection and protein measurement, ethnicity, education level, household income, smoking, alcohol consumption, BMI, and the first 20 genetic PCs. The dashed lines represent Bonferroni correction ($P < 0.05 / (2920 \times 2)$) and FDR correction ($q < 0.05$) when considering social isolation and loneliness simultaneously. The pie charts depict the proportions of identified proteins belonging to four distinct panels. **C.** 179 proteins, significant after Bonferroni correction, are plotted using ORs from the PWAS for social isolation (x-axis) against ORs from the PWAS for loneliness (y-axis). The Venn diagram shows the relationship between proteins associated with social isolation, loneliness, or both. **D.** Protein-protein interaction network for the 179 identified proteins. The node color indicates whether the protein is related to social isolation (blue) or loneliness (red). The node size reflects the modularity level estimated by the maximal clique centrality method. The line thickness represents the interaction score. Top five enriched GO biological processes and molecular functions (**E**) and KEGG pathways (**F**) for proteins related to social isolation and loneliness are presented separately. Functional enrichments that are also related to proteins associated with the other construct are marked. The dot color indicates whether the biological processes is related to social isolation (blue) or loneliness (red).

Supplementary Fig. 7 PWAS of social isolation and loneliness in Caucasians (N=35,697). **A.** Volcano plot displays the ORs (x axis) and $-\log_{10}(P\text{-values})$ (y axis) for the association of protein abundance with social isolation. **B.** Scatter plot depicting the z-statistics of the association between social isolation and proteins in random subset (x axis) alongside the z-statistics of the association between social isolation and proteins in all participants (y axis). **C.** Volcano plot displays the ORs (x axis) and $-\log_{10}(P\text{-values})$ (y axis) for the association of protein abundance with loneliness. **D.** Scatter plot depicting the z-statistics of the association between loneliness and proteins in random subset (x axis) alongside the z-statistics of the association between loneliness and proteins in all participants (y axis).

Results

“In simple models incorporating age, sex, site, technical factors, and the first 20 genetic principle components (PCs) as covariates, we found 776 proteins significantly associated with social isolation and 519 proteins associated with loneliness ($P < 0.05 / (2920 \times 2)$) (Supplementary Fig. 2). After additional adjustments for ethnicity, education level, household income, smoking, alcohol consumption, and BMI, 175 proteins associated with social isolation (Fig. 1A and Supplementary Table 1) and 26 proteins associated with loneliness (Fig. 1B and Supplementary Table 2) maintained significant at the Bonferroni-corrected threshold.” (Lines 107-114, Page 4)

“Additionally, to mitigate potential population stratification, we conducted PWAS specifically in Caucasians (N=35,697) and observed that the proteomic associative patterns were highly correlated with those found in the full sample ($r > 0.9$, Supplementary Fig. 7).” (Lines 146-148, Page 5)

“With adjustment of demographic, socioeconomical and lifestyle confounders, and the first 20 genetic PCs, our findings revealed that 90.2% of these proteins were associated with mortality,

and over 50% were linked to T2D, CVD, and stroke, whereas only 6.6% were associated with dementia (Fig. 5A and Supplementary Table 15).” (Lines 267-270, Pages 7-8)

Methods

“Demographic factors included age, sex, assessment center, ethnicity (white, mixed, Asian, black, and other), body mass index (BMI), household income (<£31000, ≥£31000), and education level.” (Lines 548-550, Page 14)

“The first 20 genetic principal components (PCs) were controlled for to mitigate potential population stratification.” (Lines 566-567, Page 14)

“Initially, a simple model was employed, controlling for age, sex, site, technical factors, and the first 20 genetic PCs.” (Lines 572-573, Page 14)

“(3) PWAS was conducted specifically in Caucasians to further mitigate the possibility of population stratification.” (Lines 581-582, Page 15)

“All models were adjusted for age, sex, ethnicity, education level, household income, smoking, alcohol consumption, BMI, and the first 20 genetic PCs.” (Lines 696-697, Page 17)

Reference

Sun, B. B., Chiou, J., Traylor, M., Benner, C., Hsu, Y. H., Richardson, T. G., ... & Whelan, C. D. (2023). Plasma proteomic associations with genetics and health in the UK Biobank. *Nature*, 622(7982), 329-338.

Limitations of MR: Authors focus primarily on the Wald ratio test, but also use the weighted median and MR Egger regression. The exposure and outcome samples overlap, which means that MR Egger regression is not appropriate for use (see Minelli et al DOI: 10.1093/ije/dyab084). As well, empirically, MR Egger is very under-powered. As an alternative to the approach taken (i.e., the methods implemented by the TwoSampleMR package), I suggest that they use the MR method developed by Zhu et al., which was specifically designed for eQTL data (https://doi.org/10.1038/ng.3538).

Response:

-- Thank you for your insightful comments regarding the limitations of MR. First, we would like to clarify that we used the inverse-variance weighted (IVW) method rather than the Wald ratio test as our primary MR analytic approach. The Wald ratio test is only applicable when there is a single instrumental variable. Additionally, as stated in the original manuscript, “GWAS summary statistics were sourced from two independent samples from the UK Biobank.” The exposure and outcome samples are non-overlapping. We have revised the Methods and Results sections to make these points clearer.

We agree that MR-Egger is known to be an under-powered method (Schmidt and Dudbridge, 2018). In response to your suggestion, we have now performed Generalized Summary-data-based Mendelian Randomization (GSMR) along with the heterogeneity in dependent

instruments (HEIDI)-outlier test (Zhu and Stephens, 2018) as an alternative approach. GSMR is a powerful MR method for testing a putative causal association between two phenotypes based on a multi-SNP model. We did not use the Summary-data-based Mendelian Randomization (SMR) as it is designed to test the effect of a single genetic variant (Zhu et al., 2016). The findings from GSMR were very consistent with the results of IVW. We have updated the Methods, Results, and Figures accordingly.

Methods

“We utilized **inverse-variance weighted (IVW)** as the primary analytic approach, with the **Wald ratio test employed if only one instrument was available.**” (Lines 652-654, Page 16)

“**To avoid bias due to participant overlap**, separate GWASs for social isolation (N=297,396) and loneliness (N=288,696) were performed using a distinct set of Caucasian participants, excluding those involved in the protein-related GWASs.” (Lines 641-644, Page 16)

“**To account for pleiotropy, three sensitivity analyses were conducted: (1) weighted median (Bowden et al., 2016), (2) MR-Egger (Bowden et al., 2015), which adds an intercept to the IVW regression to exclude confounding from uncorrelated pleiotropy, and (3) Generalized Summary-data-based Mendelian Randomization (GSMR) with heterogeneity in independent instrument (HEIDI)-outlier method (Zhu et al., 2018) to identify and remove SNPs with evidence for significant uncorrelated pleiotropy ($P < 0.01$). For GSMR, we used 10,000 randomly selected unrelated samples from the UK Biobank as a reference to determine linkage disequilibrium patterns.**” (Lines 654-660, Page 16)

Results

“GWAS summary statistics were sourced from **non-overlapping** samples from the UK Biobank.” (Lines 229-230, Page 7)

“Results of sensitivity analyses were consistent with IVW estimates in direction and magnitude. No evidence of heterogeneity (Q-statistic, all $P > 0.3$) and horizontal pleiotropy (MR-Egger intercept, all $P > 0.3$; **HEIDI-outlier test, $P > 0.01$**) among instrument variables was detected.” (Lines 236-239, Page 7)

Figure 3 Summary of MR and colocalization analysis on the associations between loneliness and proteins and protein modules. Inverse variance weighted method was used as the primary analytic approach, complemented by weighted median, MR-Egger, and GSMR. FDR correction was applied to results for each direction and for social isolation and loneliness separately. Support for colocalization was considered strong for $PP.H4 \geq 0.8$ and medium for $0.5 < PP.H4 < 0.8$.

Reference

- Schmidt, A. F., & Dudbridge, F. (2018). Mendelian randomization with Egger pleiotropy correction and weakly informative Bayesian priors. *International journal of epidemiology*, 47(4), 1217-1228.
- Zhu, X., & Stephens, M. (2018). Large-scale genome-wide enrichment analyses identify new trait-associated genes and pathways across 31 human phenotypes. *Nature communications*, 9(1), 4361.
- Zhu, Z., Zhang, F., Hu, H., Bakshi, A., Robinson, M. R., Powell, J. E., ... & Yang, J. (2016). Integration of summary data from GWAS and eQTL studies predicts complex trait gene targets. *Nature genetics*, 48(5), 481-487.
- Bowden, J., Davey Smith, G., Haycock, P. C., & Burgess, S. (2016). Consistent estimation in Mendelian randomization with some invalid instruments using a weighted median estimator. *Genetic epidemiology*, 40(4), 304-314.
- Bowden, J., Davey Smith, G., & Burgess, S. (2015). Mendelian randomization with invalid instruments: effect estimation and bias detection through Egger regression. *International journal of epidemiology*, 44(2), 512-525.

Authors performed analysis of cis and trans pQTLs. Trans-pQTLs require some additional analytic considerations: MR assumes that instruments influence the outcome only through modifying the exposure variable. Yet, Sun et al. suggest that trans-pQTLs effects influence multiple domains (<https://doi.org/10.1038/s41586-023-06592-6>). This increases the possibility that the results of trans-pQTL MR were confounded by correlated horizontal pleiotropy. While authors somewhat considered uncorrelated horizontal pleiotropy, they did not consider

correlated horizontal pleiotropy, as detailed by Morrison et al (DOI: 10.1038/s41588-020-0631-4). For trans-pQTL analyses, I suggest the authors also attempt to implement CAUSE MR analyses or LHC-MR analyses (<https://doi.org/10.1038/s41467-021-26970-w>).

Response:

-- Thank you for the helpful comment. Horizontal pleiotropy is indeed a significant consideration, occurring when a variant affects both the exposure and the outcome through either uncorrelated or correlated mechanisms. In our study, we used weighted median, MR-Egger regression, and the HEIDI-outlier test to address uncorrelated pleiotropy. However, we did not initially account for correlated pleiotropy, which is a limitation.

Following your recommendation, we conducted Causal Analysis Using Summary Effect estimates (CAUSE) (Morrison et al., 2020) to account for both correlated and uncorrelated pleiotropy for significant MR results. The CAUSE analysis did not show a significant difference between the sharing model and the causal model (Bayesian model comparison: GFRA1, $P=0.53$; ADM, $P=0.96$; FABP4, $P=0.999$; TNFRSF10A, $P=0.24$; ASGR1, $P=0.47$). Since CAUSE is the only method capable of distinguishing causality from correlated pleiotropy, we cannot completely rule out the possibility of horizontal pleiotropy.

Despite this, we found consistent evidence for causal associations between loneliness and plasma proteins using IVW, weighted median, and GSMR methods. Additionally, colocalization analysis strongly supported the colocalization between loneliness and two proteins, ADM and ASGR1, together providing compelling evidence of a causal effect of loneliness on these plasma proteins. However, caution is warranted in interpreting these findings due to the CAUSE results. Further investigations in larger, more powerful datasets are necessary to determine whether the observed associations are due to causality rather than pleiotropy.

We have introduced the CAUSE method in the Methods, summarized the findings in the Results section, and provided the detailed CAUSE results in Supplementary Table 12. This limitation has also been discussed in the Discussion section.

Methods

“For significant MR results, Causal Analysis Using Summary Effect estimates (CAUSE) (Morrison et al., 2020) was performed with a large set of LD-pruned SNPs ($r^2 < 0.01$ with an arbitrary $P \leq 1 \times 10^{-3}$) to further consider correlated pleiotropy.” (Lines 660-663, Page 16)

Results

“However, CAUSE could not distinguish a model of causality from correlated pleiotropy for the five proteins which showed significant causal links with loneliness using IVW (Supplementary Table 12).” (Lines 240-242, Page 7)

Discussion

“Interestingly, we identified significant causal impacts only from loneliness to **five** proteins, and the robustness of these findings was confirmed through sensitivity analyses **except the CAUSE model.**” (Lines 364-366, Page 10)

“**Fourthly, consistent causal effects of loneliness on five proteins were inferred using different methods except for CAUSE, indicating the possibility of correlated pleiotropy. Correlated pleiotropy is common and may frequently contribute to false-positives in MR studies (Morrison et al., 2020). Further investigations in larger, more powerful datasets will be needed to determine whether the observed associations are due to causality rather than pleiotropy.**” (Lines 436-441, Page 11)

Reference

Morrison, J., Knoblauch, N., Marcus, J. H., Stephens, M., & He, X. (2020). Mendelian randomization accounting for correlated and uncorrelated pleiotropic effects using genome-wide summary statistics. *Nature genetics*, 52(7), 740-747.

Protein networks limitation: How sensitive were the protein network results to the selection of parameters in the Netboost analysis? It would be concerning if major changes happened based on minor perturbations of parameter choice, which seemed arbitrary (authors can correct me if I am wrong about this).

Response:

-- Thank you for your comment. The parameters in our Netboost analysis align with those from a prior large-scale network-based proteomic study on dementia (Walker et al., 2023). Specifically, we set the minimum module size to 20 and applied a Spearman filter method with a soft power of $\beta=2$. Using these settings, we identified protein modules significantly associated with SI/LO enriched in distinct biological functions. For example, protein module M4 showed enrichment in immune-related pathways, M8 in metabolic processes, M12 in neutrophil degranulation, and M3 in complement systems.

Evaluation of predicted modules is challenging, as there is no ground truth of ‘correct’ modules in molecular networks; hence, biological interpretability usually serves as a crucial criterion (Choobdar et al., 2019). Additionally, assessments from a comprehensive module identification competition indicate no significant correlation between module size and performance score (Choobdar et al., 2019).

To ensure robustness, we explored the impact of different soft power settings ($\beta=3$ and $\beta = \text{null}$, set automatically based on the scale-free topology criterion) on our results. We found consistency among the networks identified across different parameter settings. Specifically, a clear cluster of modules associated with SI and LO were identified ($\beta=2$: M4, M8, and M12; $\beta=3$: M1, M3, M6, M10, M13, and M15; $\beta=\text{null}$: M2, M3, and M5) by an unsupervised hierarchical cluster analysis. Additionally, the modules consistently showed enrichment in similar biological functions.

In conclusion, our findings from the protein co-expression network analysis demonstrate robustness across different parameter settings. We have updated the Methods and Results sections accordingly and included comparative results of parameter settings in Supplementary Fig. 16-18.

Methods

“Consistent with prior network-based proteomic analyses (Walker et al., 2023), we set the minimum module size to 20, applied a Spearman filter method with a soft power of $\beta=2$, and an “unsigned” network approach in clustering. **Additionally, we compared protein networks constructed with different soft powers (3 and null) against these parameters.**” (Lines 611-614, Page 15)

Results

“**Sensitivity analysis demonstrated that the biological relevance of the identified modules associated with social isolation and loneliness remained robust across various co-expression network construction parameters (Supplementary Fig. 16-18).**” (Lines 215-218, Page 6)

Supplementary Fig. 16 Comparison between protein co-expression networks constructed using different parameters. A. Heat map showing the association between module eigengenes identified by a soft power of $\beta=2$ (row) and $\beta=3$ (column). **B.** Heat map showing the association between module eigengenes identified by a soft power of $\beta=2$ (row) and $\beta=\text{null}$ (column). An unsupervised hierarchical cluster analysis was used to group modules based on their associations.

Supplementary Fig. 17 Association of protein co-expression network, constructed by soft power of $\beta=3$, with social isolation and loneliness. Association of module expression with social isolation (A) and loneliness (B). Enrichment analysis results for the proteins of modules 1 (C), 3 (D), 6 (E), 13 (F), and 15 (G), respectively. Top three significantly enriched pathways from each database are displayed.

Supplementary Fig. 18 Association of protein co-expression network, constructed by soft power of β =null, with social isolation and loneliness. Association of module expression with social isolation (A) and loneliness (B). Enrichment analysis results for the proteins of modules 2 (C), 5 (D), and 6 (E), respectively. Top three significantly enriched pathways from each database are displayed.

Reference

- Walker, K. A., Chen, J., Shi, L., Yang, Y., Fornage, M., Zhou, L., ... & Coresh, J. (2023). Proteomics analysis of plasma from middle-aged adults identifies protein markers of dementia risk in later life. *Science translational medicine*, 15(705), eadf5681.
- Choobdar, S., Ahsen, M. E., Crawford, J., Tomasoni, M., Fang, T., Lamparter, D., ... & Marbach, D. (2019). Assessment of network module identification across complex diseases. *Nature methods*, 16(9), 843-852.

Other limitations: Why did authors choose to examine social isolation and loneliness as binary measures? It seems like there would be a loss of statistical power from dichotomizing these phenotypes, rather than assessing them on a continuum. It would be a shame if this variable selection was only due to the statistical complexity involved (i.e. due to available software, that it is easier to model them as binary variables in logistic regression rather than as ordinal ones).

Response:

-- In our study, we used a multi-item approach to assess SI and LO. SI was defined by three questions of living alone, social contact and participation in social activities, each of which was

dichotomously recoded and summed to yield a total score with a range of 0 to 3. LO was measured by two items (“do you feel lonely” and “how often are you able to confide in someone close to you”), ranging from 0 to 2. We subsequently recoded SI and LO into dichotomous variables, aligning with their common use in previous studies (Elovainio et al., 2017; Hakulinen et al., 2018). This is largely due to their positively skewed distribution and the limited number of individuals scoring at the higher end.

We acknowledge your concern regarding the potential loss of statistical power and agree that dichotomization may not capture the full variability of these constructs. Therefore, we performed sensitivity analyses using the raw scores of SI and LO, employing ordered logistic regression (using the ‘polr’ function in R) while controlling for all covariates from the primary model. These analyses identified 580 proteins associated with SI and 125 with LO at a significance level of $P < 0.05 / (2920 \times 2)$. The number of significant proteins associated with SI and LO using raw scores was three and five times greater, respectively, compared to dichotomous variables. Notably, GDF15, the most significant protein associated with SI, remained the top protein in the ordered logistic regression, and PCSK9, the protein most significantly associated with LO, ranked third in the ordered logistic regression. 170 out of 175 proteins associated with SI, and 23 out of 26 proteins associated with LO identified by the logistic regression remained significant in the ordered logistic regression. Therefore, the results indicate that using raw scores and ordered logistic regression yield results very consistent with those obtained using dichotomous variables and logistic regression but with greater statistical power. Furthermore, we treated SI and LO as continuous variables and modeled them using linear regression, yielding results consistent with those from ordered logistic regression.

We retained the analysis with dichotomous variables as the primary results in the manuscript to facilitate additional analyses, such as the four-category analysis suggested by Reviewer #3. We have included the results of ordered logistic regression in our sensitivity analyses and presented them in Supplementary Fig. 5, with details outlined in the Methods section of the manuscript.

Results

“First, ordered logistic regressions using raw scores of social isolation and loneliness were performed to test the robustness and potential dose-dependent associations with plasma protein levels. This approach yielded results consistent with those from dichotomous variables and logistic regression but with greater statistical power (580 and 125 proteins associated with SI and LO, respectively; Supplementary Fig. 5).” (Lines 136-140, Pages 4-5)

Methods

“(1) To address the limitations of dichotomous variables potentially losing statistical power, raw scores were used and modeled with ordered logistic regression, controlling for all covariates in the primary model.” (Lines 577-579, Page 15)

Figure. Distribution of SI and LO raw scores.

Supplementary Fig. 5 PWAS of social isolation and loneliness using ordered logistic regression. A. Volcano plot displays the Betas (x axis) and $-\log_{10}(P\text{-values})$ (y axis) for the association of protein abundance with social isolation. **B.** Volcano plot displays the Betas (x axis) and $-\log_{10}(P\text{-values})$ (y axis) for the association of protein abundance with loneliness. The dashed line represents Bonferroni correction ($P < 0.05 / (2920 \times 2)$). **C.** Betas for proteins significantly associated with social isolation (x axis) or loneliness (y axis). **D.** Venn diagram of significant proteins for social isolation and loneliness.

Figure. PWAS of social isolation and loneliness using linear regression.

Reference

Elovainio M, Hakulinen C, Pulkki-Råback L, et al. Contribution of risk factors to excess mortality in isolated and lonely individuals: an analysis of data from the UK Biobank cohort study. *Lancet Public Health*. 2017; 2(6): e260-e266.

Hakulinen C, Pulkki-Råback L, Virtanen M, Jokela M, Kivimäki M, Elovainio M. Social isolation and loneliness as risk factors for myocardial infarction, stroke and mortality: UK Biobank cohort study of 479 054 men and women. *Heart*. 2018; 104(18): 1536-1542.

Reviewer #3:

Remarks to the Author:

Shen et al. used a proteomics approach to examine the proteins associated with social isolation and loneliness, the causal roles of social isolation/loneliness on these proteins, and the mediation effects of the proteins between social isolation/loneliness and diseases/mortality. This is a comprehensive, well-executed, and well-written study. It addresses an important area of research – how social risk factors “get under the skin”. I do not have any major concerns about the methods, results, and inference. The comments below are mainly for clarification, but I think addressing these comments will help the paper further. I do think there is an error with the figure numbering in the main text though. It should be corrected in the revision.

Response:

-- Thank you for the positive feedback and thorough review, which significantly enhance the manuscript. We apologize for the error in figure numbering and have addressed each of the following comments point-by-point.

Abstract

1. Does the “MR-identified proteins” refer to the 8 proteins or the 2 proteins (ADM and ASGR1). If it is the 8 proteins, I think it would help to list them.

Response:

-- Thank you for the suggestion. We have revised the abstract as:

“These MR-identified proteins (GFRA1, ADM, FABP4, TNFRSF10A, and ASGR1) exhibited broad associations with other blood biomarkers, as well as volumes in brain regions involved in interoception, emotional, and social processes.” (Lines 42-45, Page 2)

Introduction

1. Line 80 – 82: why did the author investigate the associations of the MR-identified proteins with brain volume specifically? Why not measurement of physical function?

Response:

-- Thank you very much for the question. The social brain hypothesis posits that a brain system evolved to support the social complexity, important for living in a society (Dunbar and Shultz, 2007). There have been an increasing number of neuroimaging studies which investigate the neurobiology of social isolation and loneliness (Zoyetti et al., 2021). For example, our previous work identified that social isolation is related to lower volumes in temporal, frontal and other (e.g., hippocampal) regions, partly mediating the association between social isolation and cognitive function (Shen et al., 2022). Therefore, we hypothesize that proteins causally related to social isolation and/or loneliness may also show relationship with brain structures responsible for social processing.

Additionally, we investigated the associations of the MR-identified proteins with 229 blood biomarkers, including blood biochemistry, blood count, and metabolics. These biomarkers represent broad physical functions such as liver, renal, bone, hormone, and inflammation.

To ensure our analyses are reasonable, we have now briefly stated the logic of relating proteins and brain volumes in the Introduction section, as follows:

“To explore the relationship between the MR-identified proteins and broad physical functions, we investigated their associations with other blood biomarkers. Based on the social brain hypothesis (Dunbar and Shultz, 2007) and increasing research on the neurobiology of social isolation and loneliness (Zoyetti et al., 2021), we further related these proteins to brain volumes.” (Lines 81-85, Page 3)

Reference

Dunbar, R. I., & Shultz, S. (2007). Evolution in the social brain. *science*, 317(5843), 1344-1347.
Zoyetti, N., Rossetti, M. G., Perlini, C., Brambilla, P., & Bellani, M. (2021). Neuroimaging studies exploring the neural basis of social isolation. *Epidemiology and Psychiatric Sciences*, 30, e29.

Shen, C., Rolls, E. T., Cheng, W., Kang, J., Dong, G., Xie, C., ... & Feng, J. (2022). Associations of social isolation and loneliness with later dementia. *Neurology*, 99(2), e164-e175.

Results

1. It is hard to follow the sample sizes for different analyses. It is stated at the beginning of this section that the study population included 42,204 participants. However, the results for GDF15 and PCSK9 specified N = 41,534, which is also different from the sample size for CXCL14. Was the sample size different because different people had missing values for different proteins, e.g., due to quality control? It could help if the authors state the reason.

Response:

-- We apologize for any confusion caused by the description of the sample sizes. We started with 53,026 participants who had baseline proteomic data after quality control (QC) by the UK Biobank. Among them, 49,251 participants had complete SI and LO data. In the primary model, we controlled for age, sex, batch, time gap between blood collection and protein measurement, ethnicity, education level, current smoking status, alcohol consumption, BMI, household income, site, and the first 20 genetic PCs. Consequently, we ultimately included 42,062 participants who had data for proteins, SI, LO, and all covariates. However, due to the QC, different participants had missing values for different proteins. As a result, the sample size for individual proteins ranged from 30,783 to 41,401, and only 3,016 participants had data for all proteins. Since PWAS is performed independently for each protein, to increase statistical power, we included those participants with the corresponding protein data.

Now we have provided a flow chart of participant selection in Supplementary Fig. 1 and detailed the specific sample sizes for significant proteins in Supplementary Table 1 and 2 of the PWAS results. Additionally, we have revised the Methods and Results sections to clarify the sample sizes description.

Methods:

“Due to the QC process, different participants had missing values for different proteins, resulting in varying sample sizes for individual proteins.” (Lines 487-488, Page 12)

Results

“Our primary study population included 42,062 participants (56.4±8.2 years and 52.3% female) from the UK Biobank, who had quality-controlled proteomic data and complete behavioral data including social isolation, loneliness, and all covariates. A flow chart of participant selection is shown in Supplementary Fig. 1.” (Lines 95-98, Page 4)

Supplementary Fig. 1 Flow chart of participant selection. UKB-PPP Consortium pre-selected participants were based on the Data-Field 30903. Participants from the COVID-19 repeat-imaging study were based on the Data-Field 41000.

2. Continuing with the last comment, there are other sample sizes that need clarification. The sample sizes for social isolation and loneliness were different. The numbers under the “Proteomic measurements” section do not add up. 46,595 participants at baseline, 6,376 individuals pre-selected by the UKB-PPP consortium, and 1,268 individuals participating in the COVID-19 repeat-imaging study at multiple visits do not add up to 54,219 participants from whom blood plasma samples were collected. Then, the sensitivity analysis “exclusively within the randomly selected subset (N=36,467)” does not match with any of the numbers above. I think it will help if a flow chart is added to the supplement figure connecting all the sample sizes and statement the reasons for exclusion.

Response:

-- Thank you for the valuable suggestion. We have now provided a flow chart (Supplementary Fig. 1) and revised all incorrect or inconsistent sections.

Methods

“The UK Biobank Pharma Proteomics Project (UKB-PPP) consortium conducted proteomic profiling on blood plasma samples collected from 53,026 participants at baseline, using the Olink™ Explore 3072 Proximity Extension Assay between April 2021 and February 2022. The sub-cohort comprised a randomly selected subset of 45,507 participants at baseline, 6,229 individuals pre-selected by the UKB-PPP consortium, and 1,290 individuals participating in the COVID-19 repeat-imaging study at multiple visits.” (Lines 473-478, Page 12)

Results

“Given the reported sampling bias in the UKB-PPP sub-cohort composition, we replicated the primary analyses specifically within the randomly selected subset (N=36,250), which is highly representative of the overall UK Biobank population.” (Lines 141-143, Page 5)

3. Line 146-148: state at the end of the sentence the sample size since it is different from the main analysis. I think it is important because sample size could have been a factor why the number of significant proteins reduced.

Response:

-- Thank you for the suggestion. 3,284 out of 42,062 participants lacked depressive symptoms, which might reduce the statistical power, especially the estimation of *P*-values. We have now provided the sample size of the sensitivity analysis in the main manuscript:

Results

“Considering the established link between loneliness and depression, we explored the potential influence of depressive symptoms on the proteomic association of social isolation and loneliness (N=38,778).” (Lines 160-162, Page 5)

4. Sensitivity analysis additionally adjusted for depression: I wonder why Supplementary Fig. 7A and 7B did not use the same beta-beta plots as in Supp. Fig. 4 B and D. I cannot reconcile the statement in the main text that the adjustment attenuated the associations and the lack of shift in the z-statistics distributions in Supp. Fig. 7A and 7B. I also think that another volcano plot can help here, in case a different set of proteins were significant after additional adjustment of depression. That's the overlap of the z-score distribution in A and B was due to associations for some proteins being strengthened. The same comment applies to Supp. Fig. 8.

Response:

-- According to your suggestions, we have now revised the figures of PWAS results with adjustment for depression (Supplementary Fig. 10), physical activity (Supplementary Fig. 11), and social isolation or loneliness (Supplementary Fig. 12). The figures include volcano plots and z-z plots, which show: (1) Significant proteins associated with social isolation and loneliness after multiple comparison correction; (2) Correlation of the proteomic associative patterns with or without the specific covariate.

Supplementary Fig. 10 PWAS of social isolation and loneliness accounting for depressive symptoms (N=38,778). **A.** Volcano plot displays the ORs (x axis) and $-\log_{10}(P)$ -values (y axis) for the association of protein abundance with social isolation. **B.** Scatter plot depicting the z-statistics of the association between social isolation and proteins in the primary model (x axis) alongside the z-statistics of the association between social isolation and proteins further controlling for depressive symptoms (y axis). **C.** Volcano plot displays the ORs (x axis) and $-\log_{10}(P)$ -values (y axis) for the association of protein abundance with loneliness. **D.** Scatter plot depicting the z-statistics of the association between loneliness and proteins in the primary model (x axis) alongside the z-statistics of the association between loneliness and proteins further controlling for depressive symptoms (y axis). **E.** The alteration in ORs for the association between proteins and loneliness was assessed with or without adjustment for depressive symptoms.

Supplementary Fig. 11 PWAS of social isolation and loneliness accounting for physical activity (N=34,548). **A.** Volcano plot displays the ORs (x axis) and $-\log_{10}(P)$ -values (y axis) for the association of protein abundance with social isolation. **B.** Scatter plot depicting the z-statistics of the association between social isolation and proteins in the primary model (x axis) alongside the z-statistics of the association between social isolation and proteins further controlling for total MET minutes (y axis). **C.** Volcano plot displays the ORs (x axis) and $-\log_{10}(P)$ -values (y axis) for the association of protein abundance with loneliness. **D.** Scatter plot depicting the z-statistics of the association between loneliness and proteins in the primary model (x axis) alongside the z-statistics of the association between loneliness and proteins further controlling for total MET minutes (y axis).

Supplementary Fig. 12 PWAS incorporating both social isolation and loneliness (N=42,062). **A.** Volcano plot displays the ORs (x axis) and $-\log_{10}(P)$ -values (y axis) for the association of protein abundance with social isolation. **B.** Scatter plot depicting the z-statistics of the association between social isolation and proteins in the model 2 (x axis) alongside the z-statistics of the association between social isolation and proteins further controlling for loneliness (y axis). **C.** Volcano plot displays the ORs (x axis) and $-\log_{10}(P)$ -values (y axis) for the association of protein abundance with loneliness. **D.** Scatter plot depicting the z-statistics of the association between loneliness and proteins in the model 2 (x axis) alongside the z-statistics of the association between loneliness and proteins further controlling for social isolation (y axis).

5. I wonder if the authors have considered creating 4 categories of social isolation and loneliness, i.e., socially isolated but not lonely (these people may just enjoy being alone), not socially isolated but lonely, socially isolated and lonely, and not isolated and not lonely. Would such intersections give more insights to the underlying construct of impoverished social relationships?

Response:

-- Thank you for the insightful suggestion. To further investigate the underlying construct of impoverished social relationships, we categorized participants into four groups: neither isolated nor lonely (SI-LO-; N=36,100), socially isolated but not lonely (SI+LO-; N=3,273), not isolated but lonely (SI-LO+; N=2,057), and socially isolated and lonely (SI+LO+; N=632). Multinomial logistic regression was performed with all covariates in the primary model. Compared with the SI-LO- group, we found 116 proteins differed significantly in the SI+LO- group, eight in the SI-LO+ group, and 22 in the SI+LO+ group. Interesting, GDF15 was identified as the top differentiated protein in SI+LO- and SI+LO+ groups; PCSK9 was the top differentiated protein in the SI-LO+ group.

We have reported the results in the sensitivity analysis results section and Supplementary Fig. 13, and briefly described the analysis in the Methods section.

Results

“To further investigate the underlying construct of impoverished social relationships, multinomial logistic regressions were conducted to test the association between the four-group classification and plasma protein levels. Compared with participants who were neither isolated nor lonely (N=36,100), 116 proteins differed significantly in the socially isolated but not lonely group (SI+LO-; N=3,273), eight in the not isolated but lonely group (SI-LO+; N=2,057), and 22 in the socially isolated and lonely group (SI+LO+; N=632) (Supplementary Fig. 13). Interesting, GDF15 was identified as the top differentiated protein in SI+LO- and SI+LO+; PCSK9 was the top differentiated protein in SI-LO+.” (Lines 170-178, Page 5)

Methods

“(6) Participants were categorized into four groups: neither isolated nor lonely (SI-LO-), socially isolated but not lonely (SI+LO-), not isolated but lonely (SI-LO+), and socially isolated and lonely (SI+LO+). Multinomial logistic regression was performed with all covariates from the primary model, using the SI-LO- group as the reference. A likelihood ratio test was used to estimate the overall association of protein levels compared with the model with covariates only. Significant proteins related to a specific group required both a significant model fit and a significant coefficient after Bonferroni correction ($P < 0.05/2920$.” (Lines 589-595, Page 15)

Supplementary Fig. 13 PWAS of four-group classification of social isolation and loneliness (N=42,062). **A.** Volcano plot displays the ORs (x axis) and $-\log_{10}(P\text{-values})$ (y axis) for the proteins compared between [SI+LO-] and [SI-LO-]. **B.** Volcano plot displays the ORs (x axis) and $-\log_{10}(P\text{-values})$ (y axis) for the association of protein compared between [SI-LO+] and the reference. **C.** Volcano plot displays the ORs (x axis) and $-\log_{10}(P\text{-values})$ (y axis) for the association of protein compared between [SI+LO+] and the reference. **D.** Venn diagram of significant proteins for three groups compared with the reference.

6. I think the main figure numbering was off. The main figure starts with Figure 2 in the text but with Figure 1 in the figures.

Response:

-- Thank you for pointing out this mistake. We have revised the figure and table numbering and double-checked them.

7. Line 191 – 195: I think this sentence needs some clarification. Did the fisher’s exact test test the overlap between the top 20% of proteins in each module with the significant proteins from the PWAS, or the overlap between all the proteins in a module with the significant proteins from PWAS? If it is the former, why the top 20% not all the proteins in a module?

Response:

-- We examined the relationship between significant modules identified by protein co-expression network analysis with the proteins identified by PWAS using two-sided Fisher’s exact tests. In the original manuscript, we defined hub proteins as the top 20% proteins associated with the module eigengene. Recognizing that this definition is somewhat subjective, we revised our approach to use all proteins, the top 20%, and the top 10% of proteins in each module. The results remained consistent. After Bonferroni correction, both protein sets associated with social isolation and loneliness were significantly enriched in M4, and the results were consistent across all proteins, the top 20%, and the top 10% of proteins in each module.

We have now revised the Results and Methods sections, as well as providing detailed results in Supplementary Table 9.

Results

“Moreover, we examined the relationship between significant modules with the proteins identified by PWAS using two-sided Fisher’s exact tests. After Bonferroni correction, both protein sets associated with social isolation and loneliness were significantly enriched in M4, and the results were consistent across all proteins, the top 20%, and the top 10% of proteins in each module (Supplementary Table 9). It’s noteworthy that both approaches—the PWAS and the protein co-expression networks—were consistent and complementary, together providing more comprehensive information.” (Lines 219-225, Pages 6-7)

Methods

“To assess the consistency between the identified protein networks and the proteins associated with social isolation or loneliness identified through PWAS, **two-sided** Fisher’s exact tests were conducted to determine whether the identified proteins were significantly enriched in the identified networks **using all proteins in the module, as well as the top 20% and top 10% of proteins that exhibit the most significant correlation with the corresponding module eigengene.**” (Lines 616-621, Pages 15-16)

8. Line 195 – 197: I wonder if the conclusion of consistent finding was solely based on the overlap between the proteins in the module and the proteins from PWAS. I wonder if the directions of association were or should be considered. For example, PCSK 9, a protein associated with elevated risk of loneliness, was part of Module 3, which showed associated with lower risk of loneliness. I recognized that the negative association of Module 3 may be driven by other proteins in the module, but this was not shown in any figures or tables.

Response:

-- Thank you for the comment. Our findings indicate that Module 3 (M3) was significantly associated with social isolation (OR=0.90, 95% CI 0.87 to 0.94, $P=9.7 \times 10^{-7}$) and loneliness (OR=0.93, 95% CI 0.89 to 0.98, $P=5.5 \times 10^{-3}$). This suggests that higher scores on M3 are linked to lower risks of social isolation and loneliness. Additionally, all proteins within M3 exhibited a negative correlation with the module eigengene. For instance, PCSK9 showed a correlation coefficient of -0.19, indicating that higher PCSK9 levels were associated with lower M3 eigengene and increased risks of social isolation and loneliness. Therefore, the results from our protein co-expression network analysis align with those from PWAS analysis. We have now provided proteins in significant modules and correlation to the module eigengene in Supplementary Tables 5-8.

9. I am confused about the mediation analysis. Why were the PERM kept in the paper if there are obvious limitations and another method is better. Cannot the PERM be estimated using natural indirect effect / (natural indirect effect + natural direct effect) from the counterfactual-based analysis (which the authors calculated)? If I am not mistaken, the first approach assumes no exposure-mediator interaction whereas the second approach makes no such assumption. It is weird to me that the PERM was estimated with that assumption and the significance was determined without that assumption (Figure 5 based on figure numbering).

Response:

-- Thank you for the comment. In our study, we employed two approaches to investigate the mediation effect of proteins on the relationship between loneliness and health outcomes. The first approach, which is straightforward and widely used in epidemiological research (Lu et al, 2013; Etemadi et al., 2017; Elovainio et al., 2017; Wang et al., 2023), involves comparing hazard ratios (HR) from two Cox proportional hazards models: one adjusted for confounders only, and the other adjusted for both confounders and mediators. The difference in HRs between these models is used to estimate the percentage of excess risk mediated (PERM) (Lin, Fleming, and Gruttola, 1997). However, as you pointed out, this approach lacks causal assumption, and the uncertainty of PERM limits its statistical inference.

To address this limitation, we also reported the mediation proportion using the counterfactual-based analysis, which calculates the natural indirect effect divided by the sum of the natural indirect effect and the natural direct effect. Our findings showed that the mediation proportions estimated by both methods were very similar, thereby supporting the reliability of our results.

Based on your suggestion, we have revised Figure 5C to show the mediation proportion estimated by the counterfactual-based analysis (represented by dot size) to avoid confusion. We have also provided detailed results of the mediation estimates from both methods in the supplementary tables (Supplementary Tables 16 and 17) and clarified the analysis logic in the Methods section.

Results

“Although this approach provided a quantifiable contribution of proteins and is widely used in epidemiological research (Lu et al., 2013; Etemadi et al., 2017), the statistical significance and causal interpretation were unclear.” (Lines 285-287, Page 8)

Methods

“To support the reliability of our results, we used two approaches to estimate the contribution of proteins explaining the relationship between social isolation/loneliness and health outcomes.” (Lines 700-701, Page 17)

Figure 5 Associations between identified proteins and morbidity and mortality. C. Mediation analysis of MR-identified proteins on the connection between loneliness and six health outcomes using counterfactual-based analysis. Age and sex were included as confounders. The dot size and color represent the proportion of mediation estimated by (indirect effect/(indirect effect + direct effect)). Circles indicate significance after Bonferroni correction ($P < 0.05/(5 \times 6)$) and asterisks denote significance after FDR correction ($q < 0.05$) when considering all tested proteins simultaneously.

Reference

- Lu, Y., Hajifathalian, K., Ezzati, M., Woodward, M., Rimm, E. B., & Danaei, G. (2013). Metabolic mediators of the effects of body-mass index, overweight, and obesity on coronary heart disease and stroke: a pooled analysis of 97 prospective cohorts with 1·8 million participants. *Lancet*, 383(9921), 970-983.
- Etemadi, A., Sinha, R., Ward, M. H., Graubard, B. I., Inoue-Choi, M., Dawsey, S. M., & Abnet, C. C. (2017). Mortality from different causes associated with meat, heme iron, nitrates, and nitrites in the NIH-AARP Diet and Health Study: population based cohort study. *bmj*, 357.
- Elovainio, M., Hakulinen, C., Pulkki-Råback, L., Virtanen, M., Josefsson, K., Jokela, M., ... & Kivimäki, M. (2017). Contribution of risk factors to excess mortality in isolated and lonely individuals: an analysis of data from the UK Biobank cohort study. *The Lancet Public Health*, 2(6), e260-e266.
- Wang, Y. X., Sun, Y., Missmer, S. A., Rexrode, K. M., Roberts, A. L., Chavarro, J. E., & Rich-Edwards, J. W. (2023). Association of early life physical and sexual abuse with premature mortality among female nurses: prospective cohort study. *bmj*, 381.

Lin, D. Y., Fleming, T. R., & De Gruttola, V. (1997). Estimating the proportion of treatment effect explained by a surrogate marker. *Statistics in medicine*, 16(13), 1515-1527.

Methods

1. It is stated that the sensitivity analysis for sex and age was performed independently in the stratified groups, so I assumed that no interaction terms were used in the statistical model. If that's the case, can you elaborate further how exactly the p-values were obtained, e.g., what test or statistics were used?

Response:

-- Logistic regression was used to investigate the association between social isolation and plasma protein levels, as well as loneliness and proteins, separately for males (N=20,076) and females (N=21,986). Covariates include age, site, batch, time gap between blood collection and protein measurement, ethnicity, education level, household income, smoking, alcohol consumption, BMI, and the first 20 genetic PCs. Interaction terms were not included in the analyses. Two-sided *p*-values were used to test significance. We have now revised the Methods section to make it clear.

“(4) Subgroup analyses were undertaken to examine potential sex (male vs female) and age (<60 vs ≥60 years) differences. **Logistic regression with covariates from the primary model was performed for separate groups**, and the statistical significance of group differences was evaluated using z score differences.” (Lines 582-585, Page 15)

2. Two sample MR for Module 3: How was SNPs selected for Module 3? Was it SNPs associated with any proteins in Module 3? It is not described in Methods.

Response:

-- Thank you for the question. Protein module expression was quantified by taking the first principal component score for the rank matrix of each protein module (module eigengene). We performed GWASs on protein modules significantly associated with social isolation/loneliness, using Caucasian participants with QC-ed genotyping data from the UK Biobank. We have now revised the Methods to clarify this process:

“Genome-wide association studies (GWASs) on plasma protein abundance, **using protein level for individual proteins and module eigengene for protein networks**, were conducted in Caucasian participants with **QC-ed** genotyping data from the UK Biobank.” (Lines 638-641, Page 16)

3. Did I miss it or there is no method description for the associations of the proteins and blood biomarkers and brain volumes?

Response:

-- We have now added a section titled “Associations to other blood biomarkers and brain volumes” in the Methods, as follows:

“Associations to other blood biomarkers and brain volumes

Linear regression was used to investigate the associations between proteins and protein modules related to social isolation/loneliness and various blood biochemical, haematological, and metabolic markers, as well as cortical and subcortical volumes. The same covariates as in the primary model were controlled for. Additionally, intracranial volume (ICV) and the time gap between baseline and imaging collection were considered in neuroimaging analyses.” (Lines 684-689, Page 17)

4. The counterfactual-based method (ref 34) requires a model of the exposure given confounders and a model of the probability densities of continuous mediator (proteins and protein modules) given the exposure and confounders. Please describe the specifications of these models. This is important information as the validity of the mediation analysis relies on the correct specification of these models (ref 34).

Response: Thank you for the helpful suggestion. We have now revised the Methods as follows:

“In this study, we employed a straightforward procedure based on marginal structural models that directly parameterize the natural direct and indirect effects of interest (Lin, Fleming, and Gruttola, 1997). **Based on the results of MR analysis, we found a significant causal relationship between loneliness and protein levels. Therefore, loneliness was used as the exposure, protein level as the mediator, and disease and mortality as the outcome, with age and sex as confounders.** The Aalen additive hazard model (Aalen, 1980) was utilized for the survival outcome. Confidence intervals were computed as the estimate of the natural direct or indirect effect plus/minus 1.96 times a robust standard error.” (Lines 706-713, Pages 17-18)

Reference

- Lin, D. Y., Fleming, T. R., & De Gruttola, V. (1997). Estimating the proportion of treatment effect explained by a surrogate marker. *Statistics in medicine*, 16(13), 1515-1527.
- Aalen, O. (1980, February). A model for nonparametric regression analysis of counting processes. In *Mathematical Statistics and Probability Theory: Proceedings, Sixth International Conference, Wisla (Poland), 1978* (pp. 1-25). New York, NY: Springer New York.

Discussion

1. Well-written and reasonable. I wonder if authors can discuss the possibility that GDF15 may be dysregulated as a result of underlying disease pathology (hence the strong associations in the Cox models even before clinical manifestation of the diseases) but no causal indication in the MR. GDF15 was not found to be causally associated with diseases including hypertension, diabetes, heart disease, stroke, and cancer. [1] Also, my understanding is that GDF15 has an anti-inflammatory function [2], yet GDF15 was associated with higher risk of all diseases. What do the authors think about this counterintuitive findings?

*[1] Tanaka T, Basisty N, Fantoni G, et al. Plasma proteomic biomarker signature of age predicts health and life span. *Elife*. 2020;9:e61073. Published 2020 Nov 19. doi:10.7554/eLife.61073*

[2] Pence BD. Growth Differentiation Factor-15 in Immunity and Aging. *Front Aging*. 2022 Feb 9;3:837575. doi: 10.3389/fragi.2022.837575. PMID: 35821815; PMCID: PMC9261309.

Response:

-- Thank you for your insightful comment and question. In our study, GDF15 is identified as the top protein related to social isolation, the 11th strongest protein related to loneliness, and the top-risk protein for CVD, stroke, dementia, depression, and mortality. However, we did not find any causal relationships between GDF15 and social isolation or loneliness. A possible explanation is that the increases in GDF15 observed in pathologies might be a consequence of these diseases, potentially resulting from inflammation caused by these diseases, rather than a cause (Luan et al., 2019). It is also possible that associations with disease development are potentially compensatory (Wang et al., 2021). Indeed, GDF15 appears to have broad and diverse functional roles in various conditions. While greater expression of GDF15 occurs in pathologies, it also exerts anti-inflammatory effects, potentially playing a protective role (Pence, 2022). Additionally, both mice and human studies have shown that metformin and exercise increase circulating levels of GDF15 (Kleinert et al., 2018; Day et al., 2019). Therefore, the disparate roles of GDF15 might be mediated by different mechanisms that are not fully understood.

We have discussed these issues as follows:

“Remarkably, GDF15 emerged as the top-risk protein for CVD, stroke, dementia, depression, and mortality. A recent study identified GDF15 plasma levels as the top robust predictor across 14 categories of disorders and all-cause mortality (You et al., 2023). However, GDF15 might also exert anti-inflammatory effects by inhibiting macrophage activation, thus potentially playing a protective role (Pence, 2022). The disparate roles of GDF15 might be mediated by different mechanisms that are not fully understood. Additionally, consistent with our results, GDF15 was not found to be causally associated with diseases such as hypertension, diabetes, heart disease, stroke, and cancer (Tanaka et al., 2020). Therefore, the increases in GDF15 observed in pathologies might be a consequence of these diseases, potentially resulting from inflammation caused by these diseases, rather than a cause (Luan et al., 2019). It is also possible that associations with disease development are potentially compensatory (Wang et al., 2021). These pieces of evidence may imply that GDF15 is not specific to a particular disease, yet this does not rule out its potential mechanistic relevance to neuroinflammation or other processes pertinent to the link between social relationships and health.” (Lines 405-418, Page 11)

Reference

- Luan, H. H., Wang, A., Hilliard, B. K., Carvalho, F., Rosen, C. E., Ahasic, A. M., ... & Medzhitov, R. (2019). GDF15 is an inflammation-induced central mediator of tissue tolerance. *Cell*, 178(5), 1231-1244.
- Wang, D., Day, E. A., Townsend, L. K., Djordjevic, D., Jørgensen, S. B., & Steinberg, G. R. (2021). GDF15: emerging biology and therapeutic applications for obesity and cardiometabolic disease. *Nature Reviews Endocrinology*, 17(10), 592-607.
- Pence, B. D. (2022). Growth differentiation factor-15 in immunity and aging. *Frontiers in Aging*, 3, 837575.
- Kleinert, M., Clemmensen, C., Sjøberg, K. A., Carl, C. S., Jeppesen, J. F., Wojtaszewski, J. F., ... & Richter, E. A. (2018). Exercise increases circulating GDF15 in humans. *Molecular metabolism*, 9, 187-191.
- Day, E. A., Ford, R. J., Smith, B. K., Mohammadi-Shemirani, P., Morrow, M. R., Gutgesell, R. M., ... & Steinberg, G. R. (2019). Metformin-induced increases in GDF15 are important for suppressing appetite and promoting weight loss. *Nature Metabolism*, 1(12), 1202-1208.
- You, J., Guo, Y., Zhang, Y., Kang, J. J., Wang, L. B., Feng, J. F., ... & Yu, J. T. (2023). Plasma proteomic profiles predict individual future health risk. *Nature Communications*, 14(1), 7817.

We thank you for your excellent and helpful comments which we feel have strengthened the manuscript in its revised form. We accordingly trust that this revised manuscript is now suitable for publication in *Nature Human Behaviour* and we look forward to hearing from you.

Yours sincerely,

Professor Barbara Sahakian

Professor Jianfeng Feng

10 July 2024

on behalf of all authors, who have approved the revision

Nature Human Behaviour, Manuscript #: NATHUMBEHAV-24010198A

“Plasma proteomic signatures of social isolation and loneliness: Associations to morbidity and mortality” by Chun Shen, Ruohan Zhang, Jintai Yu, Barbara J. Sahakian, Wei Cheng, and Jianfeng Feng

Thank you for the very helpful and constructive comments on this paper. We have carefully revised the paper according to the suggestions received and trust that the paper will soon be accepted for publication. Your suggestions made are shown below in *blue and italics*, and our responses in normal font preceded by --. Changes made to the paper for the revision are shown below within “...”, and in **red** font in the revised paper. The scripts used in this study has been updated and can be found at https://github.com/chunshen617/Proteomics_loneliness.

REVIEWER COMMENTS:

Reviewer #1:

Remarks to the Author:

The authors performed all additional analysis I suggested and made necessary changes in the manuscript.

Response:

-- Thank you for your positive feedback and the time and attention you have devoted to evaluating our work.

Reviewer #3:

Remarks to the Author:

The authors addressed most of my comments, but I have follow-up comments. These two comments should be addressed before the publication of the paper:

Response:

-- Thank you for acknowledging our efforts to address your concerns. We appreciate your additional questions, which have further strengthened the paper. Below, we provide a comprehensive point-by-point response to all of the comments.

1. The authors confirmed that the sensitivity analysis for sex and age was performed independently in the stratified groups and interaction terms with sex or age were not included in the models. But it is still unclear how the p-value in Line 152-154 “Nevertheless, three proteins (fatty acid-binding protein 4 [FABP4], adrenomedullin [ADM], and GDF15; $P < 0.05/(175+26)$) showed a more pronounced association with loneliness in males than in females (Supplementary Fig. 8C and 8D).” The authors’ response mentioned that “two-sided p-values were used to test significance” but for which term was the p-value calculated? Why did the author choose not to use the interaction term with sex or age which is more common approach to test effect modification? I could not find the test of the sex and age difference in the codes provided.

Response:

-- Thank you for your insightful comment. We apologize for the confusion. In our original analysis, we conducted logistic regressions separately for males and females. To examine the differences in the relationship between protein levels and social isolation or loneliness across sexes, we calculated the *z*-score difference between the protein coefficients from these regressions. This *z*-score difference was then converted into a *p*-value, which was compared to a significance threshold of $0.05/(175+26)$, accounting for the total number of significant proteins identified from the PWAS across all samples. We briefly described this approach in the original manuscript: "the statistical significance of group differences was evaluated using *z* score differences."

We agree with your suggestion that including interaction terms is a more standard approach for testing potential moderation effects of sex and age. In response, we have now added interaction terms (i.e., sex×protein and age×protein, separately) in the logistic models for the entire sample. The results indicated that neither the sex interaction (social isolation-related proteins, $p>0.03$; loneliness-related proteins, $p>0.08$) nor the age interaction (social isolation-related proteins, $p>0.03$; loneliness-related proteins, $p>0.001$) reached statistical significance after multiple comparison correction. These findings suggest that the relationship between social isolation, loneliness, and plasma proteins is minimally influenced by sex and age.

Additionally, we have included the PWAS results of the subgroup analyses in Supplementary Figures 8 and 9 to provide more detailed information on potential sex and age differences. We have also updated the code to clarify the analysis process. The Methods and Results sections have been revised as follows:

Results

"Next, interaction terms between sex or age and protein levels were included in logistic models to assess potential sex and age differences. No significant interaction effects between sex or age and proteins associated with social isolation or loneliness were observed ($P>0.05/(175+26)$). The PWAS results for males (N=20,076) and females (N=21,986) are shown in Supplementary Fig. 8. Additionally, the PWAS results for younger (<60, N=24,171) and older (≥ 60 , N=17,891) groups are presented in Supplementary Fig. 9." (Lines 149-154, Page 5)

Methods

"Sex and age differences were evaluated by incorporating interaction terms between sex or age and protein levels into logistic models, adjusting for the same covariates as in the primary model. Additionally, subgroup PWAS analyses were conducted separately for males and females, as well as for younger (<60 years) and older (≥ 60 years) individuals, to provide further insights." (Lines 579-583, Page 15)

Supplementary Fig. 8 PWAS of social isolation and loneliness in male (N=20,076) and female (N=21,986), respectively. Covariates include age, site, batch, time gap between blood collection and protein measurement, ethnicity, education level, household income, smoking, alcohol consumption, BMI, and the first 20 genetic PCs. **A.** Volcano plot displays the ORs (x axis) and $-\log_{10}(P)$ -values (y axis) for the association of protein abundance with social isolation in males. **B.** Volcano plot displays the ORs (x axis) and $-\log_{10}(P)$ -values (y axis) for the association of protein abundance with social isolation in females. **C.** Volcano plot displays the ORs (x axis) and $-\log_{10}(P)$ -values (y axis) for the association of protein abundance with loneliness in males. **D.** Volcano plot displays the ORs (x axis) and $-\log_{10}(P)$ -values (y axis) for the association of protein abundance with social isolation in females. **E.** Scatter plot depicting the z-statistics of the association between social isolation and proteins in females (x axis) alongside the z-statistics of the association between social isolation and proteins in males (y axis). Red points indicate the 175 identified proteins significantly related to social isolation in the overall population. **F.** Scatter plot depicting the z-statistics of the association between loneliness and proteins in females (x axis) alongside the z-statistics of

the association between loneliness and proteins in males (y axis). Red points indicate the 26 identified proteins significantly related to loneliness in the overall population.

Supplementary Fig. 9 PWAS of social isolation and loneliness in older (≥ 60 years, $N=17,891$) and younger (< 60 years, $N=24,171$), respectively. Covariates include age, sex, site, batch, time gap between blood collection and protein measurement, ethnicity, education level, household income, smoking, alcohol consumption, BMI, and the first 20 genetic PCs. **A.** Volcano plot displays the ORs (x axis) and $-\log_{10}(P)$ -values (y axis) for the association of protein abundance with social isolation in older group. **B.** Volcano plot displays the ORs (x axis) and $-\log_{10}(P)$ -values (y axis) for the association of protein abundance with social isolation in younger group. **C.** Volcano plot displays the ORs (x axis) and $-\log_{10}(P)$ -values (y axis) for the association of protein abundance with loneliness in older group. **D.** Volcano plot displays the ORs (x axis) and $-\log_{10}(P)$ -values (y axis) for the association of protein abundance with social isolation in younger group. **E.** Scatter plot depicting the z-statistics of the association between social isolation and proteins in younger group (x axis) alongside the z-statistics of the association between social isolation and proteins in older group (y axis).

(y axis). Red points indicate the 175 identified proteins significantly related to social isolation in the overall population. F. Scatter plot depicting the z-statistics of the association between loneliness and proteins in younger group (x axis) alongside the z-statistics of the association between loneliness and proteins in older group (y axis). Red points indicate the 26 identified proteins significantly related to loneliness in the overall population.

2. Based on the authors' response, the models for the counterfactual-based mediation analysis only adjusted for age and sex for both mediator (protein) models and outcome (disease and mortality) models. The codes confirmed this. However, this is inconsistent with the PWAS analysis which adjusted a lot more covariates. If those covariates in the PWAS were considered confounders for loneliness-protein associations, why were they not confounders anymore in the mediation analysis? Going back to my original comments, the counterfactual-based method requires the outcome model and mediator models to be correctly specified. Including only age and sex is unlikely to satisfy this requirement. At least, the authors should include all the covariates used in the PWAS for the protein models and consider whether these covariates are confounder for the outcome models.

Response:

-- Thank you for your valuable comment. The counterfactual-based method indeed assumes that no unmeasured confounders are present in the exposure-outcome, mediator-outcome, and exposure-mediator relationships (Lange, Vansteelandt, and Bekaert, 2012). Based on your insightful suggestion, we have revised both the mediator and outcome models to better satisfy this assumption.

In the mediator model, we now adjust for the same covariates used in the PWAS, including age, sex, site, batch, time gap between blood collection and protein measurement, ethnicity, education level, household income, smoking, alcohol consumption, BMI, and the first 20 genetic PCs. In the outcome model, we controlled for all covariates except protein technical factors and genetic PCs. Additionally, we applied the same covariate adjustments in the Cox-based mediation analyses as in the outcome model of the counterfactual-based method to ensure consistency between the two approaches.

As a result of these revisions, loneliness was no longer significantly related to the incidence of type 2 diabetes (HR=1.10-1.13, $P>0.18$; sample size varied across proteins). Consequently, we excluded type 2 diabetes from the counterfactual-based mediation analysis. Although the mediating effect of proteins on the relationship between loneliness and health outcomes was attenuated, it remained statistically significant.

We have updated the Results and Methods sections accordingly, revised Fig. 5C, and provided details in Supplementary Tables 16 and 17. The corresponding code has also been updated to reflect these changes.

Results

“After controlling for demographic, socioeconomic and lifestyle confounders, loneliness was not associated with the incidence of T2D and was therefore excluded from further mediation analyses. The largest effects were observed for mortality (PERM 8.6%-16.3%) and CVD (PERM 5.6%-8.3%) (Supplementary Table 16). Notably, ADM emerged as the primary mediator linking loneliness to various health outcomes, including CVD (8.3%), dementia (4.4%), stroke (7.8%), and mortality (16.3%). Although this approach provided a quantifiable contribution of proteins and is widely used in epidemiological research, the statistical significance and causal interpretation were unclear. Consequently, we used a counterfactual-based mediation analysis to assess the direct and indirect effects in the relationship between loneliness and health outcomes, through the MR-identified proteins. All five proteins significantly mediated the association between loneliness and CVD, stroke, and mortality after Bonferroni correction (Fig.5C and Supplementary Table 17). The estimated proportion of the indirect effect to the total effect was comparable to the PERM results. Interesting, ADM was the only protein that significantly mediated the relationship between loneliness and all four diseases (CVD: indirect effect= 9.0×10^{-5} , 95% CI 7×10^{-5} to 1.1×10^{-4} , $P < 1 \times 10^{-16}$; dementia: indirect effect= 2.7×10^{-5} , 95% CI 1.6×10^{-5} to 3.7×10^{-5} , $P = 1.4 \times 10^{-6}$; depression: indirect effect= 1.8×10^{-5} , 95% CI 7.1×10^{-6} to 2.9×10^{-5} , $P = 0.001$; stroke: indirect effect= 3.5×10^{-5} , 95% CI 2.5×10^{-5} to 4.6×10^{-5} , $P = 5.4 \times 10^{-11}$) and mortality (indirect effect= 2.6×10^{-4} , 95% CI 2.3×10^{-4} to 2.9×10^{-4} , $P < 1 \times 10^{-16}$).” (Lines 278-296, Page 8)

Methods

“First, we estimated the percentage of excess risk mediated (PERM) using two Cox proportional hazards models, with the formula $[\text{HR}_{(\text{age, sex, site, ethnicity, education level, household income, smoking, alcohol consumption, and BMI adjusted})} - \text{HR}_{(\text{age, sex, site, ethnicity, education level, household income, smoking, alcohol consumption, BMI, and protein adjusted})}] / [\text{HR}_{(\text{age, sex, site, ethnicity, education level, household income, smoking, alcohol consumption, and BMI adjusted})} - 1] \times 100$.” (Lines 702-706, Page 17)

“Therefore, loneliness was treated as the exposure, and protein levels were used as the mediator. In the mediator model, we included all the covariates used in the PWAS (i.e., age, sex, site, batch, time gap between blood collection and protein measurement, ethnicity, education level, household income, smoking, alcohol consumption, BMI, and the first 20 genetic PCs). In the outcome model, all covariates except protein technical factors and genetic PCs were adjusted for.” (Lines 711-716, Page 18)

Fig 5. Associations between identified proteins and morbidity and mortality. **A.** Associations between proteins and protein modules related to social isolation or loneliness with the incidence of five diseases and mortality were examined using Cox proportional hazard models. Covariates included age, sex, ethnicity, education level, household income, smoking, alcohol consumption, BMI, and the first 20 genetic PCs. The top three proteins most strongly associated with the diseases and mortality were labeled. The dot color indicates a risk (red), protective (blue), or non-significant (grey) correlation. The dot size represents the absolute difference between HR and 1. The dashed line represents the Bonferroni correction ($P < 0.05 / (183 \times 6)$). **B.** HRs and 95% CIs for the MR-identified proteins are presented, with asterisks indicating significance after Bonferroni correction in (A). **C.** Mediation analysis of MR-identified proteins on the connection between loneliness and five health outcomes using counterfactual-based analysis. The dot size and color represent the proportion of mediation estimated by (indirect effect / (indirect effect + direct effect)). Asterisks indicate significance after Bonferroni correction ($P < 0.05 / (5 \times 5)$).

Reference

Lange, T., Vansteelandt, S., & Bekaert, M. (2012). A simple unified approach for estimating natural direct and indirect effects. *American journal of epidemiology*, 176(3), 190-195.